# Thin-film lithium niobate terahertz differential field detectors with a bandwidth reaching 3 terahertz

Alexa Herter [1] ✉, Amirhassan Shams-Ansari[2,3], Marko Lončar [2] & Jérôme Faist [1] ✉

Broadband and sensitive detection of terahertz (THz) radiation is critical for advances in fields like telecommunications, spectroscopy, and quantum physics. We present a compact and high-performance THz field detector based on resonant THz antennas printed along near-infrared waveguides on thin-film lithium-niobate. These antennas were shown to have their peak response between 250 GHz to 1 THz, depending on their geometry, while the non-resonant nature of the interaction enables THz detection to be achieved up to 3 THz. We show that combining two such antennas in an integrated Mach Zehnder interferometer allows for a measurement of the discrete time derivative of the THz waveform, while using a single antenna measures the instantaneous derivative of the field. Using this approach, we have achieved a noise equivalent intra-cavity field as low as 1.9 Vm$^{-1}$ for an integration time of 100 ms, corresponding to a single-shot noise-equivalent field of 4.6 kV m$^1$, using a pulsed laser operating at 1575 nm with 76 μW average power. Our device would enable the next generation of compact detectors for applications in spectroscopy and quantum optics.

In recent decades, the spectral range of terahertz (THz) radiation has emerged as a rapidly growing area of research due to its unique properties and diverse applications. The development of THz technology has implications across a range of fields, including security, telecommunications[1] and quantum physics[2,3]. Furthermore, THz radiation is of particular value for its applications in spectroscopy and imaging[4]. Many elementary excitations, such as electrons[5], spins[6], or molecular vibrations[7], as well as the band gaps of superconducting materials[8,9], occur at energies corresponding to THz frequencies. For imaging, THz waves provide non-invasive insight into biological tissues[10] and security scanning.

The relatively low frequency of THz radiation compared to optical light allows femtosecond laser pulses to resolve THz fields on sub-cycle timescales and generate single-cycle THz pulses. This enables direct access to the electric field of THz waves using time-domain techniques based on ultrafast carrier dynamics in photoconductive antennas or the interaction between THz and near-infrared fields inside a nonlinear crystal[11–15]. The sub-cycle resolution offered by these techniques provides access to both the amplitude and phase of the THz electric field, an essential requirement for coherent spectroscopy and quantum sensing applications. In particular, detection based on nonlinear optics directly detects the electric field of vacuum fluctuations, and the possibility of accessing both quadratures offers a new experimental window into quantum electrodynamics[2,3,16–19]. While such systems have traditionally been bulky and alignment-sensitive, advances in integrated nonlinear optics now offer compact, stable, and scalable possibilities, opening the door to on-chip THz quantum technologies.

Recent THz sensors using a silicon-on-insulator Mach-Zehnder interferometer combined with nonlinear polymers have shown excellent performance levels[20,21]. The integrated waveguides direct near-infrared probe pulses into the plasmonic gap of THz antennas filled

[1]ETH Zurich, Institute of Quantum Electronics, Zurich, Switzerland. [2]Harvard John A. Paulson School of Engineering and Applied Sciences, Harvard University, Cambridge, MA, USA. [3]DRS Daylight Solutions, San Diego, CA, USA. ✉e-mail: aherter@ethz.ch; jfaist@ethz.ch

with the nonlinear polymer, enhancing the electro-optic coupling through the double confinement of the near-infrared and THz signal and the large optical nonlinearity of the polymer. However, challenges such as optical losses and polymer degradation limit probe power and device longevity.

Progress in thin-film lithium niobate technology has enabled the development of miniaturized and broadband spectrometers operating in the near-infrared[22], as well as high-speed electro-optic modulators with demonstrated bandwidths up to 300 GHz and projections exceeding 500 GHz[23–25]. Although the modulator bandwidth is ultimately constrained by the speed of electronic readout, the material itself supports much higher frequencies. The efficient nonlinear interaction between near-infrared pulses and THz fields has been demonstrated by generating THz waveforms combining integrated waveguide structures and THz antennas on thin-film lithium niobate[26,27]. Furthermore, it has been shown that THz detection using thin-film lithium niobate is feasible even in the absence of resonant structures[28], implying that the incorporation of antennas should significantly improve sensitivity through enhanced field confinement and interaction strength.

In this work, we demonstrate that resonant terahertz antennas combined with Mach-Zehnder interferometers on thin-film lithium niobate waveguides enable highly sensitive detection of broadband THz fields. In contrast to a recent report employing a similar platform[29], we use a single resonant antenna per interferometer arm instead of a pair of geometrical arrays, each consisting of up to nine antennas. This design choice enables the detection of THz waveforms with bandwidths up to 3 THz. In addition to the interferometric waveguide-based detection scheme, we investigate a second approach that is sensitive to the temporal derivative of the THz field. We provide a detailed analysis of the underlying detection principles, spectral response, and noise characteristics to assess the potential of THz detection on thin-film lithium niobate for applications in spectroscopy and quantum sensing.

The core of the device principle comprises printed gold THz antennas positioned along near-infrared waveguides to confine the free-space THz field and ensure efficient interaction with the near-infrared probe signal propagating through the integrated device structure. The Pockels effect inside the lithium niobate waveguide induces a phase shift in the near-infrared probe signal, which is then converted into an intensity modulation either within an integrated interferometer structure or by using a spectral discriminator.

To obtain an efficient intensity modulation through a Mach-Zehnder interferometer, the phase-shift induced in each of the two arms has to be in opposite directions. Although previous work utilizing silicon photonics has employed opposite poling of the nonlinear material[3,20], recently a new approach on thin-film lithium niobate has been demonstrated, which circumvents this additional fabrication step by using spatially shifted antennas[29]. By shifting the antennas along the waveguide, a temporal shift of the THz field is exploited with respect to the probe pulse entering the interaction region, resulting in an opposite direction of the electric field at the moment of the interaction. We demonstrate that the time-domain signal effectively yields the difference of the THz waveform at two distinct points in time.

Additionally, we perform a THz field derivative measurement scheme that leverages the finite optical bandwidth and, therefore, time resolution of the probe pulses. We show that the nonlinear interaction in a single antenna induces a spectral shift in the probe signal proportional to the temporal derivative of the THz field. This signal can be measured directly using an optical spectral discriminator.

These two approaches not only simplify the fabrication process but also open new possibilities for quantum metrology, where knowledge of both the field and its derivative is of great interest since they constitute two conjugate counterparts[16]. Furthermore, the design flexibility of integrated THz detection allows one to tailor the spectral range and timing through antenna design and placement, offering power scalability and low power consumption for various applications.

## Results

All devices investigated in the present study are fabricated on a $x$-cut thin-film lithium niobate chip (details about fabrication and the photonic chip are in the Methods Section and the Supplementary Information of ref. 26, the design parameters varying between the devices shown in this study are listed in Table 1 of the Supplementary Information). To achieve a field-sensitive detection of THz radiation, near-infrared femtosecond pulses centered at a wavelength of 1575 nm are coupled via a grating coupler from the cleaved facet of an optical fiber into the integrated single-mode ridge waveguide. At the same time, the THz radiation is incident through the semi-insulating silicon substrate. As shown in Fig. 1, a evaporated THz gold antennas confine the THz field inside the waveguide that hosts the near-infrared optical mode. Both the THz field confined within the antenna and the near-infrared probe signal are polarized parallel to the [001]-axis, thus ensuring efficient interaction due to the large electro-optic coefficient $r_{33}$ of lithium niobate along this axis.

### Two-point difference signal with interferometric structure

For the first detection scheme, the integrated device is designed as a Mach-Zender interferometer (Fig. 1a), similar to the THz detectors based on a silicon-on-insulator platform[20,21]. A waveguide-based Y-splitter divides the incoming probe intensity $2I_0$ equally between two parallel arms and guides the near-infrared pulses through the gap of the gold antennas. The interaction region between the probe and the confined THz signal is defined by two gold electrodes placed in the center of the antenna and along the lithium niobate waveguide. The electrode gap is set to 3 μm to minimize absorption-related optical losses due to the presence of the gold electrodes.

Due to the Pockels effect, the confined THz field introduces a phase shift $\varphi_{1,2}$ on the probe pulses. This shift is proportional to the instantaneous field strength, $E_{gap}(t_0)$, at the time $t_0$ the probe crosses the antenna gap (see Fig. 1 b). Note that the frequency response of the antenna can be tailored through its geometrical characteristics and will therefore determine the actual temporal shape of the THz waveform interacting with the optical field inside the gap. The timing of the individual inter-action processes in the two interferometer arms is determined by the precise placement of the THz antennas along the waveguide as indicated by the dotted lines in Fig. 1b and d. The experimental configuration in combination with the high refractive index of the silicon substrate ensures the propagation of the THz signal inside the device perpendicular to the chip surface. As a result of the displacement ($\Delta L$) of the two antennas along the waveguides (see Fig. 1b), the phase shift $\varphi_1$ and $\varphi_2$ experienced by the two pulses is proportional to the THz field strength at two distinct times. These times are separated by the propagation delay $\delta t_{shift} = \frac{n_g \Delta L}{c}$ (Fig. 1b), where $n_g$ is the effective group refractive index of the thin-film lithium niobate waveguide and $c$ is the vacuum speed of light. At the end of the Mach-Zehnder interferometer, the two probe signals are then combined into a single waveguide, with a device-specific offset phase difference $\varphi_{MZI}$ arising due to a slight length discrepancy between the two arms. As shown schematically in Fig. 1c, an additional phase difference induced by the THz field $\varphi_1 - \varphi_2$ causes a change in the intensity transmitted through the interferometer.

By introducing a delay in the probe signal relative to the THz waveform, the system enables sampling of the difference between the components of the confined THz waveform, $E_{gap}(\tau + \delta t/2)$ and $E_{gap}(\tau - \delta t/2)$, as a function of time $\tau$. Assuming that the interaction time $\delta t_{int} = \frac{n_g l_{gap}}{c}$ within each antenna gap of length $l_{gap}$ as well as the pulse duration $t_{FWHM}$ are both much shorter than the period of the THz field, the intensity transmitted through the integrated interferometer can be expressed as:

$$I_{MZI}(\tau) = I_0 \left( 1 + \cos\left\{ \varphi_{MZI} - g_{eo} \Gamma \left[ E_{gap}\left(\tau + \frac{\delta t_{shift}}{2}\right) - E_{gap}\left(\tau - \frac{\delta t_{shift}}{2}\right) \right] \right\} \right). \quad (1)$$

In Eq. (1) the electro-optic coupling $g_{eo}$ is given by $g_{eo} = \frac{n_p^3 r_{33} \omega_0 l_{gap}}{2c}$ where $n_p = 2.137$ is the refractive index of lithium niobate at the probe

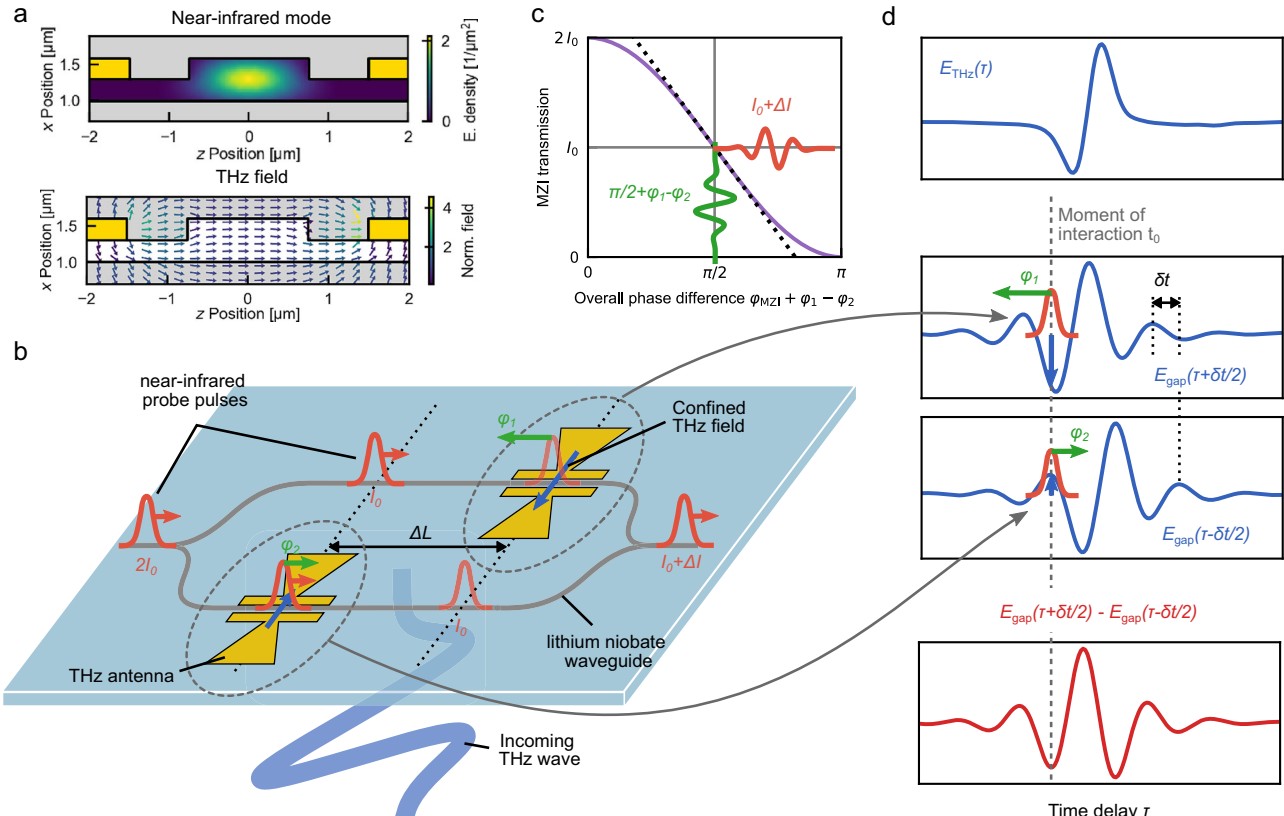

**Fig. 1 | Principle of integrated THz detection with shifted antennas in a Mach-Zehnder interferometer. a** Cross-section of the lithium niobate waveguide (color map/white) inside the antenna gap (yellow) and surrounded by the SiO₂ cladding (grey). The spatial energy density $|u_{\text{NIR}}(\mathbf{r}_\perp)|^2$ of the near-infrared probe mode simulated in a finite element simulation using COMSOL is plotted inside the lithium niobate area (upper plot). The THz field is simulated using CST Microwave Studio and is plotted as a vector field across the whole area (lower plot). **b** Thin-film lithium niobate-chip for THz detection. The intensity of the probe pulses (red) is divided equally into the two arms of an integrated Mach-Zehnder interferometer, while the THz waveform (blue) is propagating through the substrate of the chip. Two antennas on the arms of the Mach-Zehnder interferometer confine the THz field into the gap, where it overlaps with the probe pulses. After the interaction, the two probe signals are combined into one waveguide. The antennas on the two arms are displaced along the waveguide direction by a distance ΔL. **c** Interferometer

response. The transmission through a Mach-Zehnder interferometer (MZI) is plotted depending on the phase difference of the two interferometer arms (purple line). The geometric path difference of the integrated structure introduces a phase-difference of $\varphi_{\text{MZI}}$, where the slope of the transmission is maximized (black dotted line). Any additional phase difference $\varphi_1 - \varphi_2$ (green waveform) induced by the interaction with the THz field inside the antenna gaps will result in an intensity modulation $\Delta I$ (red waveform). **d** Differential waveform. Due to the displacement of the two antennas ΔL, the THz field of the probe pulses overlaps with inside the antenna gaps $E_{\text{gap}}(\tau + \frac{\delta t_{\text{shift}}}{2})$ and $E_{\text{gap}}(\tau - \frac{\delta t_{\text{shift}}}{2})$ is shifted by the propagation $\delta t_{\text{shift}} = \frac{n_g \Delta L}{c}$. The induced phases $\varphi_1$ and $\varphi_2$ are proportional to the instantaneous THz field inside the antenna in the moment of interaction (grey dotted line). According to the Mach-Zehnder interferometer response function (**c**), the temporal shape of the measured waveform $\Delta I$ (red solid line in lower plot) is given by the difference of the shifted incoming THz waveforms $E_{\text{gap}}(\tau + \frac{\delta t_{\text{shift}}}{2}) - E_{\text{gap}}(\tau - \frac{\delta t_{\text{shift}}}{2})$.

frequency, $r_{33} = 35$ pm V⁻¹ is the electro-optic coefficient of lithium niobate, $\omega_0 = 2\pi \times 191$ THz is the center angular frequency of the probe signal and $l_{\text{gap}}$ is the interaction length. The electro-optic coupling also includes a factor Γ that accounts for the spatial overlap between the THz and the probe field. By design, we aim for a device-specific phase difference between the interferometer arms of $\varphi_{\text{MZI}} = \frac{\pi}{2}$, as this results in the maximum slope of the transmitted intensity (see Fig. 1c). The waveform of the THz field confined inside the antenna gap $E_{\text{gap}}(t) = \int d\Omega \, E_{\text{THz}}(\Omega) G(\Omega) e^{-i\Omega t}$ for an incoming plane wave with a spectral field distribution $E_{\text{THz}}(\Omega)$ is determined by the field enhancement of the particular antenna design $G(\Omega)$.

While Eq. (1) highlights the key features of the detection scheme, it does not account for the effects of pulse duration and phase-matching. For a more accurate description of the electro-optic detection process, the interaction can be found in terms of a sum and difference frequency generation between the probe and the THz signal within lithium niobate with the second-order nonlinear susceptibility $\chi^{(2)}_{33}(\omega_{\text{NIR}}, \omega_{\text{NIR}}, \Omega_{\text{THz}}) = -\frac{n_{\text{NIR}}^4}{2} r_{33}$ [15]. The description in the optical frequency domain enables the direct incorporation of the probe's temporal and spectral properties via its complex spectral amplitude $\mathcal{E}_p(\omega)$.

The short interaction length within the antenna gap allows us to assume that the input fields remain unchanged while an additional nonlinear field is generated within the interaction process. As a result, the derivation remains in the linear regime in the THz field, making this approach easier to adapt to a description of quantum states of light[30]. Due to the comparatively narrow probe bandwidth of 17 nm, we assume that the group refractive index $n_g$ is constant, resulting in a device-specific offset phase shift between the two arms being constant throughout the probe pulse spectrum $\varphi_{\text{MZI}}(\omega) = \varphi_{\text{MZI}}$. Ultimately, the intensity at the end of the interferometer structure is determined as a function of the time delay $\tau$ between the probe pulse and the incoming free-space THz waveform $E_{\text{THz}}(t) = \int d\Omega \, E_{\text{THz}}(\Omega) e^{-i\Omega t}$ (derivation in Supplementary Information Note 1.1):

$$I_{\text{MZI}}(\tau) = I_0 \left(1 + \cos\varphi_{\text{MZI}} - g_{\text{eo}} \, \Gamma \, \sin\varphi_{\text{MZI}} \int d\Omega \, e^{-i\Omega\tau} R(\Omega) E_{\text{THz}}(\Omega)\right). \quad (2)$$

We find the same electro-optic coupling strength $g_{\text{eo}}$ as in Eq. (1) and the overlap factor $\Gamma = \frac{\int_{\text{LN}} |u_p(\mathbf{r}_\perp)|^2 u_{\text{THz}}(\mathbf{r}_\perp) d^2 r_\perp}{\int |u_p(\mathbf{r}_\perp)|^2 d^2 r_\perp}$ representing the spatial

mode profile of the THz $u_{\text{THz}}(\mathbf{r}_\perp)$ and the probe $u_{\text{p}}(\mathbf{r}_\perp)$ signal within the nonlinear material. Although the definition is independent of the normalization of the near-infrared mode, it is essential to define the spatial profile of the THz field in a manner that is consistent with the field enhancement of the THz antenna $G(\Omega)$ (details in Supplementary Information Note 2.1 and 2.2). By normalizing the simulated THz field to its effective mode area $\int |u_{\text{THz}}(\mathbf{r}_\perp)|^2 d^2 r_\perp = S_{\text{eff}}$, we find an overlap factor of $\Gamma = 0.54$.

As mentioned above and in accordance with Eq. (1), the change in the transmitted intensity for a given input field is optimized when the interferometer operates at the quadrature point $\varphi_{\text{MZI}} = \frac{\pi}{2}$. The spectral characteristics of the derived signal are now determined not only by the antenna's field enhancement $G(\Omega)$, but also by the frequency-dependent function filter $R(\Omega) = G(\Omega)D_{\text{shift}}(\Omega)C_{\text{p}}(\Omega)P(\Omega)$, which accounts for both the device design and the probe properties. The displacement of the antennas by the distance $\Delta L$ affects the signal through the term $D_{\text{shift}} = 2i \sin\left(\frac{\delta t_{\text{shift}}\Omega}{2}\right)$. By expressing the Sinus-term in its complex representation, the factor $D_{\text{shift}}$ under a Fourier transformation results in a difference of the field in time shifted by the propagation time $\delta t_{\text{shift}}$, analogous to equation (1). The phase-matching between the probe and the THz field is expressed by $P(\Omega) = \text{sinc}\left(\frac{\delta t_{\text{int}}}{2}\Omega\right)$, which accounts for the finite interaction time $\delta t_{\text{int}}$. Similar to bulk electro-optic sampling[30], the effect of the pulse duration is expressed in the spectral autocorrelation of the probe signal

$$C_{\text{p}}(\Omega) = \frac{\int_0^\infty d\omega\, \mathcal{E}_{\text{p}}^*(\omega)\mathcal{E}_{\text{p}}(\omega - \Omega)}{\int_0^\infty d\omega\, |\mathcal{E}_{\text{p}}(\omega)|^2}, \tag{3}$$

which gives the spectral overlap between the original complex probe spectrum $\mathcal{E}_{\text{p}}(\omega)$ and the complex spectrum $\mathcal{E}_{\text{p}}(\omega \mp \Omega)$ generated in the sum- or difference frequency generation of the probe signal with a THz field at the frequency $\Omega$. At the same time, the signal described by Eq. (2) can be converted into a signal operator and evaluated on a quantum mechanical state of the THz field by substituting $E_{\text{THz}}$ by the corresponding operator, as analogously demonstrated in ref. 30.

We evaluated the THz detection performed by integrated interferometer devices of various antenna geometries, detecting the THz emission originating from a photoconductive antenna. One of the two outputs of a dual-wavelength pulsed laser source is used to gate the photoconductive antenna, while the second output centered at 1560 nm is coupled into a lithium niobate waveguide acting as the probe signal. A delay stage controls the time delay between the near-infrared

probe signal and the THz field, which is collected and focused onto the integrated device by a pair of parabolic mirrors. By moving the stage position, we scan along the THz waveform, sampling the introduced intensity modulation of the probe signal.

The photoconductive antenna's voltage is periodically inverted and the probe intensity transmitted through the chip is measured through a lock-in amplifier, yielding directly the modulation induced by the THz field

$$\Delta I(\tau) = \gamma I_0 g_{\text{eo}} \sin\varphi_{\text{MZI}} \int d\Omega\, e^{-i\Omega\tau} R(\Omega) E_{\text{THz}}(\Omega) \tag{4}$$

on top of the background intensity $I_{\text{det}} = \gamma I_0(1 + \cos\varphi_{\text{MZI}})$. The factor $\gamma$ accounts for the total loss accumulated during the propagation from the end of the Mach-Zehnder interferometer to the photo detector. To investigate the spectral properties of different detector designs, a Fourier transformation is performed on the main pulse in the time traces detected with the respective devices. Details about the experimental setup can be found in the Methods Section and in the Supplementary Information Note 3.2. To calibrate the free-space THz signal $E_{\text{THz}}(\tau)$ incident on the lithium niobate detector, we used electro-optical sampling in a 200 μm zinc telluride crystal that measured the signal of the photoconductive antenna (details in Supplementary Information Note 3.3).

Assuming that the interferometer is operating at the quadrature point $\varphi_{\text{MZI}} = \frac{\pi}{2}$ and utilizing the electro-optic coupling constant $g_{\text{eo}}$, it is possible to translate the experimentally observed relative probe modulation into an effective spectral gap field strength

$$E_{\text{eff}}(\Omega) = \frac{\text{FT}[\Delta I(\tau)]}{g_{\text{eo}} I_{\text{det}}}. \tag{5}$$

According to equation (4), this can then be compared to the effective field modeled $E_{\text{mod}}(\Omega) = R(\Omega) \cdot E_{\text{THz}}(\Omega)$, where $E_{\text{THz}}(\Omega)$ represents the spectral field amplitude of the free space THz.

Figure 2a shows a plot of the relative intensity modulation $\frac{\Delta I(\tau)}{I_{\text{det}}}$ detected with antennas of 40 μm arm length against the time delay $\tau$ between the THz and probe signal and is compared to the predicted signals, calculated according to Eqs. (1) and (2). Both models show a good quantitative agreement with the experimental measurement. For the complete model, the probe signal is described by a Gaussian pulse with a temporal duration of $t_{\text{FWHM}} = 400$ fs. This quantity is based on the near-infrared spectrum measured after propagation through the chip and the simulated group velocity dispersion of the optical fiber and

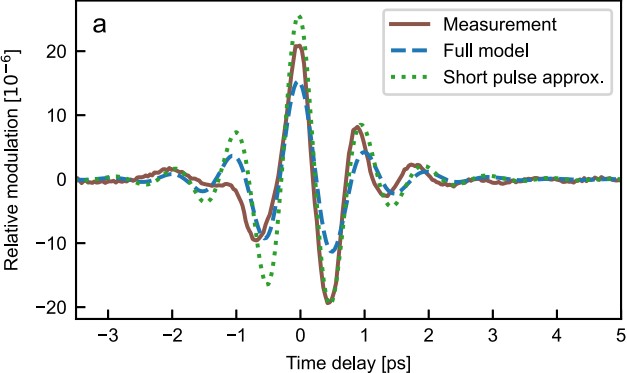

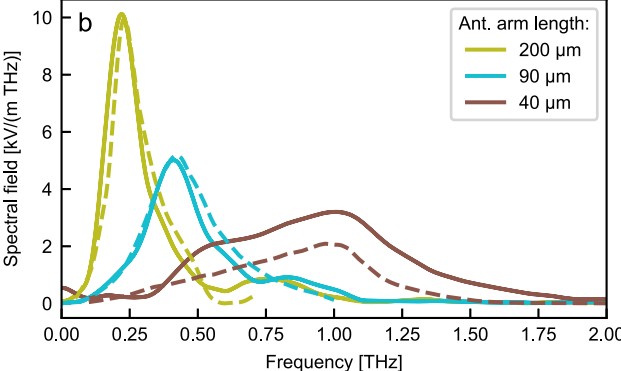

**Fig. 2 | Experimentally observed and calculated THz signals using integrated interferometer detectors. a** Time trace measured using an antenna with an arm length of 40 μm compared to the modeled signal. The measured relative transmission change $\frac{\Delta I(\tau)}{I}$ is plotted as a function of time delay $\tau$ (brown solid line) and compared to the expected signal assuming infinitely short pulses (Eq. (1), green

dotted line) and taking into account finite pulse lengths and phase-matching (Eq. (2), blue dashed line). The field incident on the antenna is 536 Vm[1] peak to peak. **b** Measured spectral effective field strength inside the antenna gap for three devices with different antenna-arm lengths 200 μm, 90 μm, and 40 μm (solid lines) compared to the calculated effective field strength using Eq. (2) (dashed lines).

integrated waveguide. The temporal shape of the derived signal is determined by field enhancement $G(\Omega)$ and therefore exhibits minimal variation when modeling according to Eq. (1) or (2) (Fig. 2a). As apparent in Fig. 2a, the description that accounts for the finite pulse duration and phase matching, expressed in a more accurate filter function $R(\Omega)$ in Eq. (2), results in a reduction of peak-to-peak modulation by a factor of 0.59 compared to the simplified model of equation (1).

The effect of the finite pulse length is reflected in the autocorrelation $C_p(\Omega_{res})$ at the resonance frequency of the antenna $\Omega_{res}$ as defined in equation (3). The value of $C_p(\Omega_{res}) = 0.54$ expresses the strong contribution of the pulse length in reducing the peak-to-peak modulation. In contrast, the factor including the phase-matching $P(\Omega_{res}) = 0.995$ for the current antenna design is close to unity and therefore hardly affects the detection efficiency. In Fig. 2a, the experimentally observed signal exceeds the theoretical prediction from the full model in Eq. (2). We attribute the highest uncertainty in these measurements to the deviation of the operation point of the Mach-Zehnder inteferometer from the ideal $\varphi_{MZI} = \frac{\pi}{2}$. In fact, the responsivity, defined as $\mathcal{R}(\Omega) = \frac{\sin \varphi_{MZI}}{1 + \cos \varphi_{MZI}} g_{eo} R(\Omega)$, and which represents the spectral relative intensity modulation per incident spectral THz field strength "increases" when $\varphi_{MZI} > \frac{\pi}{2}$. Measuring 10 different devices under the same conditions, the observed amplitude varies between 50% and 2.5 times the modeled signal. Since other uncertainties, such as the true value of the THz electric field incident on the sample or the exact temporal shape of the probe pulses, would cause a systematic deviation, we attribute the observed variation to the operation point of the interferometer. The observed deviation range corresponds to the phase-difference varying between $0.25\pi$ and $0.7\pi$ (details in the Supplementary Information Note 3.5).

In addition, we analyze the dependence of the antenna resonance frequency on the antenna geometry. A comparison between the effective field measured and modeled in the spectral domain for three bow-tie antenna designs of varying arm lengths $L_{ant}$ = 200 μm, 90 μm, and 40 μm is shown in Fig. 2b. At the respective resonance frequency of the antenna $\frac{\Omega_{res}}{2\pi}$ = 0.22 THz, 0.42 THz, and 1.03 THz, the modeled responsivity is $\mathcal{R}(\Omega_{res}) = 4.46 \times 10^6 \, mV^1, 1.56 \times 10^6 \, mV^1$, and $0.256 \times 10^6 \, mV^1$ assuming $\varphi_{MZI} = \frac{\pi}{2}$. As a result of the change in resonance frequency, the center frequency of the detected signal shifts to a higher frequency as the antenna size decreases. For the two larger antennas, there is good agreement between the simulated and detected spectral fields. However, for the 40-μm antenna, the detected signal exceeds the expected field, as already observed in the time domain and discussed above (Fig. 2a). The modulation per incident THz field and interaction length $\eta_{LN} = \frac{n_p^3 \omega_p r_{33}}{c} \cdot \Gamma \cdot G(\Omega)$ at the resonance frequency of the particular antenna is $\eta_{LN} = 80 \times 10^3 \, V^1, 29 \times 10^3 \, V^1$, and $22 \times 10^3 \, V^1$ for the three devices at 0.22 THz, 0.42 THz, and 1.03 THz, respectively. In comparison, the corresponding value for electro-optic sampling $\eta_{EOS} = \frac{n_p^3 \omega_p r_{41}}{c}$ is about two orders lower and measures $1.16 \times 10^3 \, V^1$ using zinc telluride and a probe wavelength of 800 nm or $0.23 \times 10^3 \, V^1$ in gallium arsenite with a probe signal at 1560 nm.

To illustrate the capacity of the proposed detection scheme to access frequencies above 1 THz, the Fourier transformation of the relative modulation $\frac{\Delta I(\Omega)}{I_0}$ obtained with the smallest antenna design is plotted on a logarithmic scale in Fig. 3, compared to the signal obtained from the same photoconductive antenna using state-of-the-art electro-optic sampling in zinc telluride. Due to the confinement of the THz field inside the antenna, the relative modulation at the resonance frequency of the antenna exceeds the value observed utilizing electro-optic sampling, despite the shorter interaction length of 7.5 μm compared to the 200 μm thickness of the zinc telluride crystal. The noise level, which is determined in both cases by shot-noise and therefore related to the optical power reaching the photo-detector, is higher for the integrated detector due to the lower probe power of 0.14 mW compared to 5.8 mW detecting in bulk zinc-telluride.

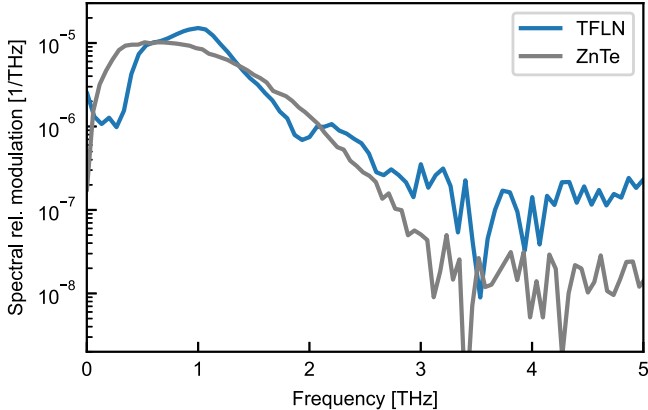

**Fig. 3 | Bandwidth of integrated THz detection.** THz spectrum (relative modulation) detected with a thin-film lithium niobate device (TFLN) optimized for 1.03 THz using an antenna arm length of 40 μm and an interaction length of 7.5 μm (blue solid line) compared to a measurement of the THz signal emitted by the same photoconductive antenna using electro-optic sampling in a 200-μm zinc telluride (ZnTe) crystal (grey solid line) in logarithmic scale.

Although the simulated antenna resonance is at 1.03 THz, the observable signal reaches 3 THz, which defines the bandwidth in the context of THz time-domain spectroscopy. The detected frequency range is comparable to the bulk electro-optic sampling technique detecting the same THz source, indicating the upper frequency limit observed in the present study is determined by the emission spectrum of the photoconductive antenna. Both high-frequency slopes of the detected spectral distributions are consistent within the experimental accuracy, thus this experiment indicates that the intrinsic cutoff of the thin-film lithium niobate detection scheme is higher than 3 THz.

Furthermore, we demonstrate the power scalability of the probe signal by detecting the photoconductive antenna emission with one device, varying the optical probe power. The resonance frequency of the utilized antenna design is at 0.37 THz with a calculated responsivity of $\mathcal{R}(\Omega_{res}) = 1.4 \times 10^6 \, mV^1$. On the photo-detector, the power range under consideration reaches from 0.26 μW to 160 μW. A reproducible THz spectrum is observed within the full power range (Fig. 4a), which indicates that nonlinear effects in the on-chip waveguide do not affect the spectral response of this device within the investigated power range. The coupling losses of about 10 dB arising at each grating coupler means that the power flowing within the integrated waveguide structure is an order of magnitude higher than the collected power. Improving the coupling between the on-chip waveguide and the photo-detector would increase the power reaching the detector, which is limited by the utilized laser source in the presented investigation, without risking limitations due to the aforementioned nonlinear effects. Figure 4b, shows that the peak-to-peak power modulations extracted from the observed time trace follow a linear behavior as expected from Eq. (2) within the whole probe power range without indicating any signs of saturation. The peak-to-peak relative modulation was averaged over the different optical power settings, resulting in a value of $9.8(3) \times 10^5$. This is slightly above the expected peak-to-peak relative modulation of $8.4 \times 10^5$, due to the uncertainty of the operation point of the Mach-Zehnder interferometer (compare Supplementary Information Note 3.5). In analogy to the noise-equivalent power of a power detector, we additionally determine the noise-equivalent field NEF $= \frac{1}{g_{eo}} \cdot \frac{\delta I_{std}}{I_0}$ by calculating the standard deviation of the detected photo-current fluctuations $\delta I_{std}$ on top of the averaged photo-current $I_0$ in the absence of any coherent THz radiation or field along the waveguide. The noise-equivalent field is defined with respect to the field within the interaction region, consistent with quantum optics conventions, where field operators are referenced to confined modes rather than incident free-space fields. Typically, the sensitivity of electro-optic detection schemes is limited by the shot-noise

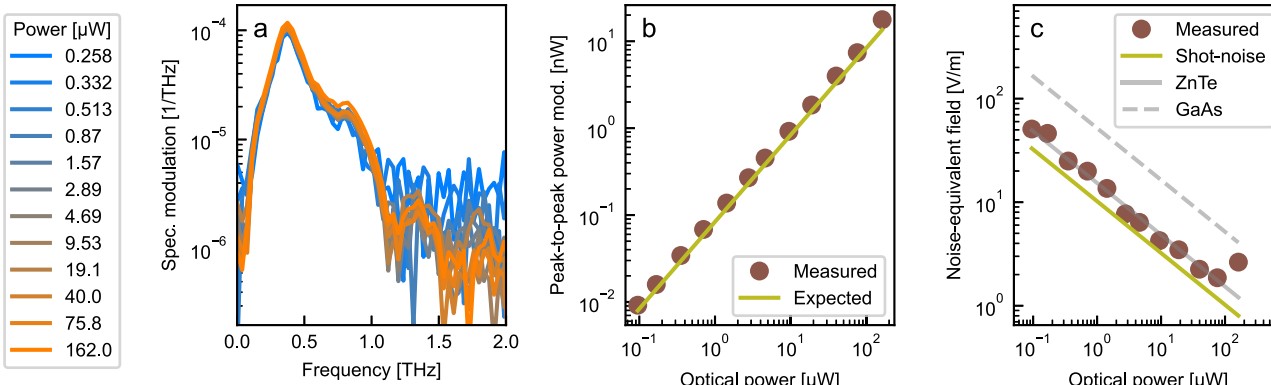

**Fig. 4 | Probe power scalability of integrated THz detection. a** Relative spectral power modulation measured using a Mach-Zehnder interferometer detector with 90 μm antenna length using probe powers between 0.258 μW and 162 μW. **b** Peak-to-peak power modulation of the probe signal using the same device, depending on the probe power reaching the photo-detector. **c** Noise-equivalent field detected with an integration time of 100 ms per point inthe absence of any THz signal compared to the expected shot-noise equivalent field plotted over the used probe power on the photo-detector. Additionally, the calculated shot-noise equivalent field for free-space electro-optic sampling using a 800 nm probe signal in a zinc-telluride crystal (ZnTe) or a 1550 nm probe signal in a gallium-arsenite crystal (GaAs), assuming the same interaction length of 45 μm.

in the photo-detectors $\frac{\delta I_{sn}}{I_0} = \sqrt{\frac{2e\Delta\nu}{I_0}}$, which is determined by the averaged induced photo-current $I_0$ and the detection bandwidth $\Delta\nu$ related to the chosen integration time of 100 ms in the current case. Figure 4c demonstrates a similar inverse square-root relationship of the expected noise equivalent field $NEF_{sn} = \frac{1}{2g_{eo}} \cdot \frac{\delta I_{sn}}{I_0}$ caused by the shot-noise within almost the full power range. The lowest noise-equivalent field of 1.9 Vm[1] is achieved using 76 μW averaged probe power. This value corresponds to a noise-equivalent field of 4.6 kV m[1] acquiring the signal at the laser's repetition rate of 60 MHz as required to study the fluctuating properties of quantum states[2,3]. The experimentally observed noise-equivalent field inside the antenna matches that of electro-optic sampling with zinc telluride, the standard material used with 800 nm probe pulses in state-of-the-art free-space electro-optic sampling, which is indicated by the solid gray line in Fig. 4c. Using gallium arsenide, which is commonly employed for electro-optic sampling using a C-band laser as in our case, the expected shot-noise equivalent field is approximately three times higher.

In comparison to the anticipated shot noise for our detection scheme, we observe a noise level that is approximately 1.5 times higher, predominantly due to the relative intensity noise of the laser and to the electric noise of the transimpedance amplifier, whose gain had to be changed as the power is scaled. Furthermore, an increase in observed signal fluctuations is evident for the highest probe power of 162 μW. At this point, the expected shot noise is 70 dB smaller than the average photo current, which exceeds the dynamic range of the transimpedance amplifier, resulting in an increased noise behavior. In the future, implementing a balanced detection scheme in the current detector design could help address this limitation. Instead of using a Y-combiner at the end of the Mach-Zehnder interferometer, a 50:50 waveguide coupler could be used to split the probe signal after the interferometer into two equal parts. These signals would then be sent to the diodes of a differential detector, so that the differential current is zero in the absence of any THz radiation. The interaction of the probe signals with an incoming THz field will lead to an imbalance of the intensity after the 50:50 waveguide coupler, which results in a differential signal similar to the approach used in advanced electro-optic sampling techniques. The combination of improved coupling and a balance detection scheme could increase the signal-over-noise ratio by a factor of up to 4.7 for the same on-chip power.

## Derivative detection with a single antenna

Finally, we present a method for detecting THz radiation with a single waveguide passing through one antenna, as an alternative to the Mach-Zehnder interferometer structure. After interacting with the THz field inside the antenna gap, the probe signal is spectrally filtered in a fiber-coupled high-pass filter and directed onto a photo-detector (Fig. 5a). As previously stated in the derivation of Eq. (1), the interaction of the near-infrared probe pulse with a THz field results in a phase shift of the near-infrared field, which is proportional to the instantaneous THz field in the limit of a short interaction length. Instead of assuming an infinitesimal short pulse resulting in a constant THz field over the pulse duration, we now incorporate the linear term of the Taylor expansion of the THz field around the center of the probe pulse $t_0$ (Fig. 5b and c). This yields a time-dependent phase shift:

$$\Delta\varphi(t) = g_{eo}\left[E_{THz}(t_0) + \partial_t E_{THz}(t_0)(t - t_0)\right], \quad (6)$$

where $\partial_t E_{THz}(t_0)$ denotes the temporal derivative of $E_{THz}$ evaluated at the time $t_0$ and $g_{eo}$ is the same electro-optic coupling strength as in Eqs. (1) and (2). Applying this time-dependent phase shift to the electric field of a Gaussian probe pulse $E_{ini}(t)$ at center frequency $\omega_0$ and Gaussian pulse width $\sigma_t$, results in a near-infrared field after the interaction of:

$$E_{int}(t) = E_{ini}(t) \cdot e^{-i\Delta\varphi(t)} = E_0 \exp\left(-\frac{(t-t_0)^2}{2\sigma_t^2}\right)e^{-i\left(\omega_0 + g_{eo}\frac{\partial E_{THz}}{\partial t}\right)(t-t_0)}e^{-ig_{eo}E_{THz}(t_0)}.$$

$$(7)$$

The interaction with the THz field causes a shift of the center frequency by $\Delta\omega = g_{eo}\partial_t E_{THz}(t_0)$, in addition to the well-known constant phase shift $\Delta\varphi_0 = g_{eo}E_{THz}(t_0)$. While an interferometer structure is necessary for the detection of the constant phase offset $\Delta\varphi_0$, the spectral shift can be identified through the use of a spectral filter. For a high (or low pass) edge filter (gray area in Fig. 5d and e) with a cutoff frequency in the center of the probe spectrum $\omega_0$, the spectral shift induced in the gap of a single antenna is translated into a modulation of the transmitted intensity (yellow area in Fig. 5d and e). In contrast to the Mach-Zehnder interferometer devices with shifted antennas, which measure the THz field in two different points in time, the single-antenna device detects the time derivative of the THz field within the time duration of the near-infrared probe pulse. In combination with a filter whose cutoff frequency is centered within the probe pulse spectrum, the temporal gradient of a THz field is converted into an intensity modulation.

To verify the theoretical prediction, we now use an integrated device consisting of a single antenna with a resonance frequency

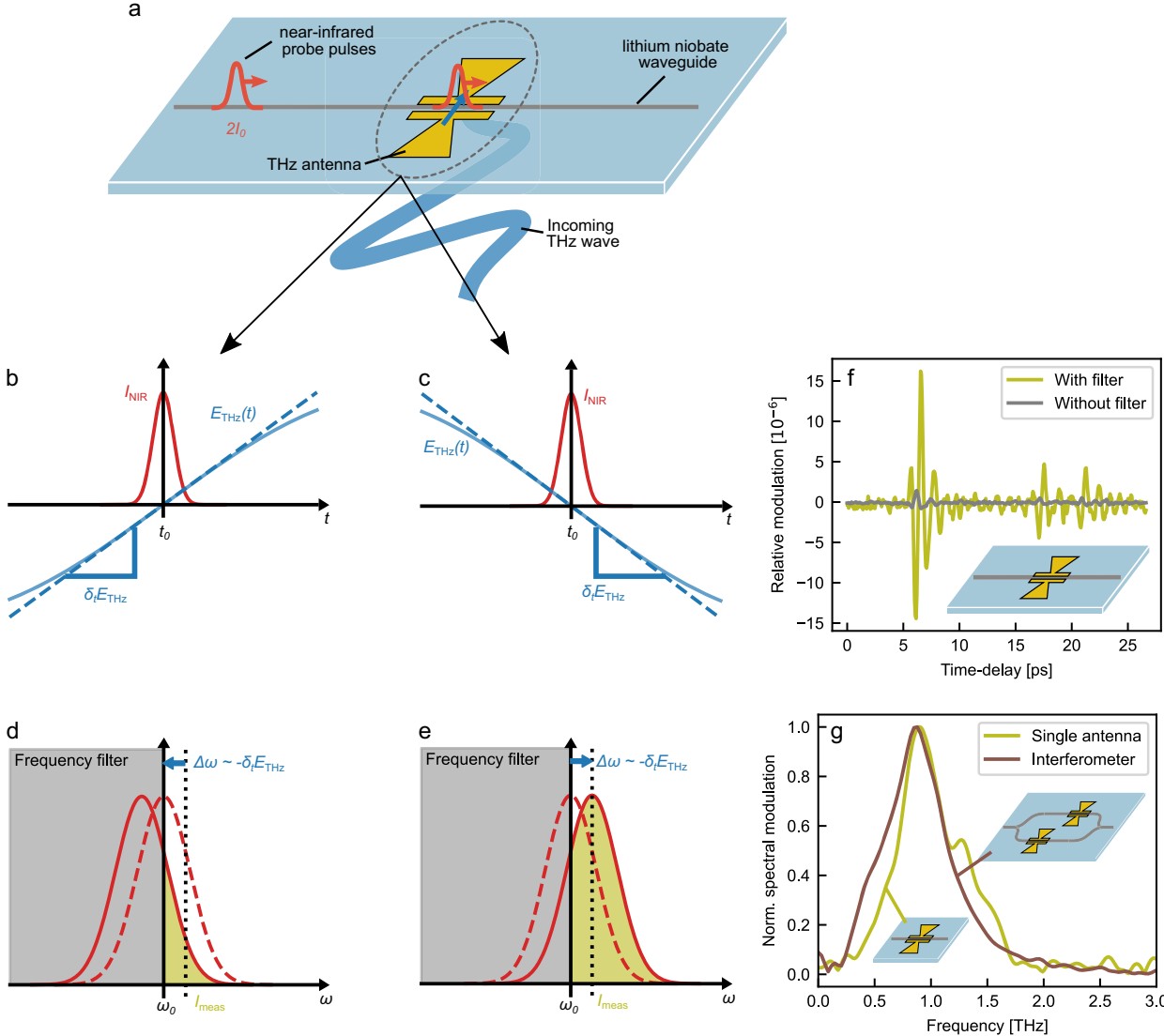

**Fig. 5 | Detection of temporal derivative based on spectral filtering of probe signal. a** Detection principle of a single-antenna device: A single near-infrared waveguide crossing through the gap a THz antenna built from the integrated structure of the device. After the probe signal propagated through the chip, it is passing through a low-pass spectral filter with a cut-off frequency $\omega_0$ in the center of the probe spectrum. **b, c** Interaction in time-domain. The THz field (blue solid line) within the temporal extent of the probe pulse is approximated in a Taylor-series up to the linear term around the center of the pulse $t_0$. The interaction is illustrated at two different delay times, once the probe pulse overlapping with a positive (**b**) and the other case a negative (**c**) slope $\partial_t E_{THz}$. d & e: Spectral shift of probe spectrum. The probe spectrum after the interaction with the THz field (red solid line) is shifted by a frequency $\delta\omega$ proportional to the gradient of the THz field $\partial_t E_{THz}$ (blue arrow) with respect to the initial spectrum (red dashed line). The spectral parts below the cut-off frequency of the high-pass filter (grey area) are suppressed, and the power reaching the detector (yellow area) depends on spectral shift $\delta\omega$. For a positive slope of the THz field (**b**) the spectrum is red-shifted and the measured intensity $I_{meas}$ is decreased (**d**), while the spectral shift and the intensity change is the inverse for a negative slope (**c, e**). **f** Measured time-trace Measured intensity change using a single antenna with a resonance frequency of 0.9 THz and a short-pass filter with a cut-off wavelength of 1575 nm (yellow solid line). In comparison, the same device is used without spectrally filtering the probe signal (grey solid line). **g** Measured THz spectrum Fourier transformation of measured intensity change (yellow solid line) compared to the spectrum observed using an interferometer device with a comparable antenna design (brown solid line).

around 0.9 THz to measure the THz emission of the previously utilized photoconductive antenna (antenna parameters in the Supplementary Information Table 1). The near-infrared signal passed through the antenna and coupled back into the optical fiber is passed through a short-pass filter with a cut-off wavelength of 1575 nm and directed onto a photo-detector (probe spectrum in Supplementary Fig. 6). The observed peak-to-peak modulation in the order of magnitude of $10^{-5}$ is comparable to the modulation observed in the Mach-Zehnder inter-ferometer structure (Fig. 5f, yellow solid line). To confirm the hypothesis that the signal is a consequence of spectral filtering, a measurement was performed without additional filtering (see Fig. 5f, gray solid line). The residual modulation was found to be more than

one order of magnitude smaller than the original signal, and it is attributable to the grating coupler, which functions as a spectral filter for the outgoing probe signal as well.

To understand the spectral properties of the single-antenna detection scheme, the normalized spectrum of the presented time-trace is compared to the signal observed with the interferometer device using shifted antennas of a similar resonance frequency in Fig. 5f.

Since the signal is caused by the temporal derivation of a field, intuitively, one expects an increasing signal for higher frequencies. At the same time, the change in intensity after the spectral filter is not linear in the induced frequency, but depends on the spectral shape of

the probe signal. In the measurement, we indeed observe a slight favoring of higher frequency components for the single antenna (yellow solid line) compared to the interferometer device (brown solid line). In both cases, the spectral shape is mainly determined by the utilized antenna resonance, so a high similarity in the spectral properties is observed.

## Discussion

In conclusion, we demonstrated broadband and sensitive THz detection on the integrated platform of thin-film lithium niobate, showcasing its significant potential for future integrated THz applications. Two distinct detection schemes were employed: the first relies on an integrated interferometer structure, while the second involves spectral filtering of the probe field after its interaction with the THz field within a single antenna. As part of the systematic investigation of the Mach-Zehnder interferometer devices, we varied the antenna arm lengths to tune the peak detection efficiency across a frequency range from 220 GHz to 1 THz. This demonstrated that the peak spectral sensitivity of the antenna can be customized for specific applications. The current antenna design incorporates short antenna gaps, enabling efficient phase-matching under the assumption of a standing THz wave within the antenna gap.

In the study of the smallest antenna optimized for detection at 1 THz, we observed a signal extending to 3 THz even with a relatively long pulse duration of approximately 400 fs, which determines the time resolution similar to other time-resolved THz detection schemes. Electro-optic sampling in bulk crystal demonstrated that the frequency limitation caused by the probe duration can be extended into the mid-infrared frequency range when using sub-10 fs pulses[31]. For a fiber-coupled system using the thin-film lithium niobate chip as shown in this study, dispersion-complementing fibers are widely available for the probe signal at the telecommunication wavelength. However, the design of lithium niobate waveguides also allows one to control the dispersion, so that, for a fully integrated solution in future development, Fourier-limited pulses in the detection region could be achieved. While the bow-tie antenna design used in the current study shows a broadband nature in the lower frequency range, different antenna designs e.g., split-ring resonators, can be used to optimize for detection at higher THz frequencies. In the current work, the comparable short interaction length ensures efficient phase matching, even when assuming a standing THz wave. At the same time, the integrated platform offers the advantage of adapting the propagation speed of the THz and probe signal by the design of the antenna geometry, the interaction region formed by the two gold bars in its gap, and the near-infrared waveguide, which has already been successfully applied to the generation of THz radiation in thin-film lithium niobate[27]. The phonon resonance at 4.5 THz limits the detection in bulk lithium niobate to below 2 THz. In contrast, the bandwidth of the integrated detection exceeds this limitation because of the low interaction of the THz radiation with the only 300 nm-thick lithium niobate layer. In fact, the usage of a lithium niobate waveguide in free-space electro-optic sampling has demonstrated a significant increase in the detection bandwidth[32]. The mentioned study predicts a detection bandwidth of up to 6.5 THz, which underlies the capacity of the THz detection on thin-film lithium niobat,e reaching a much higher bandwidth than shown so far.

Scanning the probe power over nearly three orders of magnitude from 0.26 μW to 160 μW, produced consistent THz signals in the full power range and sensitivity approaching the shot-noise limit. Our result highlights two key aspects: first, the potential for low-power THz detection, as demonstrated by successful detection at probe powers below 1 μW; and second, the current limitations in detection efficiency, which are still significantly constrained by the coupling efficiency of the grating coupler. By the use of edge coupling or replacing the fiber-based laser source with an integrated laser compatible with thin-film

lithium niobate, the probe power can be further increased, and additionally, the loss of spectral probe bandwidth at the grating couplers can be avoided. In addition to the enhanced sensitivity and reduced relative noise, the detection bandwidth of the THz frequencies can be increased because of the broader spectrum of the probe signal as opposed to grating couplers. Using a 50:50 waveguide coupler at the end of the integrated Mach-Zehnder interferometer would enable a balanced detection scheme. This can significantly improve the signal-to-noise ratio and remove the residual laser noise. Furthermore, integrating thermo-optical phase shifters on one arm of the interferometer would allow precise control of the phase difference between the arms, optimizing the device's operating point[33].

The second detection scheme investigated in the present study enables the measurement of the time derivative of the THz field using the simple integrated structure of a single antenna along a lithium niobate waveguide without any interferometer structure. Instead, a fiber-coupled spectral long-pass filter converts the phase change of the probe signal induced through the interaction with the THz field into an intensity modulation. While the signal after the interferometer devices is given by the difference of the instantaneous THz field in two different points in time, the detection scheme using only one waveguide accesses the time derivative of the THz field within the probe pulse duration. Because the spectral amplitude of the observed THz field remains unaffected by the phase shift, both techniques are equally suitable for spectroscopic applications. Furthermore, the combination of direct field measurements, leveraging opposite poling, and differential detection schemes enhances the versatility of integrated THz detection. This versatility is particularly valuable in the study of quantum states of light, where simultaneous detection of both the field and its time derivative is of great interest. The integrated platform could enable the combination of both detection schemes within a single device, paving the way for innovative applications.

Since the generation of THz radiation on thin-film lithium niobate has already been demonstrated[26,27], the present demonstration of broadband and sensitive detection completes the foundation for time-domain THz spectroscopy on the same integrated platform. In particular, the achieved detection bandwidth up to 3 THz significantly broadens the range of spectroscopically accessible phenomena compared to previously reported narrowband detection schemes[29]. This extended bandwidth enables the investigation of a wider set of elementary excitations, including low-energy vibrational, rotational, and electronic transitions occurring across the full THz spectrum. In combination with the existing toolbox of integrated components on thin-film lithium niobate[34], such as on-chip pulsed laser sources, active control of near-infrared probe power, and integrated THz transmission lines[27], the necessary technological prerequisites for realizing fully integrated THz time-domain spectroscopy are now in place.

## Methods
### Optical setup
We are using the emission of a photo-conductive antenna to characterize the detection properties of our chip-scale THz detectors in a dual-wavelength terahertz time-domain setup. A detailed sketch of the characterization setup is provided in the Supplementary Information Note 3.2. The system is based on a dual-wavelength erbium-doped fiber laser emitting femtosecond laser pulses at 1560 nm and its second harmonic at 780 nm. The signal at 780 nm is focused onto a photo-conductive antenna emitting THz radiation. A pair of parabolic mirrors collects the emitted radiation and focuses it onto the on-chip device from the substrate side. The probe signal centered at 1560 nm emitted from the cleaved facet of a single-mode fiber is coupled into the integrated waveguide via grating couplers. After propagating through the device structure and interacting with the THz signal, the probe signal is coupled back into a cleaved single-mode fiber and directed onto a sensitive photo detector. A delay stage on the 780 nm pump

signal controls the delay between the THz and the probe signal to allow sampling along the THz waveform.

## Data availability
The data generated in this study have been deposited in the Research Collection database of ETH Zurich under https://doi.org/10.3929/ethz-b-000741451.

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

## Acknowledgements
We acknowledge Ileana-Cristina Benea-Chelmus for designing the chips, fruitful discussions, and insightful comments to the manuscript. A.H. acknowledges financial support from the SNF project 207795. The fabrication of these chips was performed in part at the Center for Nanoscale Systems (CNS), a member of the National Nanotechnology Coordinated Infrastructure Network (NNCI), which is supported by the National Science Foundation under NSF Award no. 1541959. The views, opinions and/or findings expressed are those of the author and should not be interpreted as representing the official views or policies of the Department of Defense or the U.S. Government. The authors declare no competing interests.

## Author contributions
A.H. conceived the concept of detecting the THz time derivative with a single antenna. All authors developed these concepts further. A.H. built the THz time-domain fiber-coupled setup, carried out the measurements, derived the theoretical description, and performed the simulations with CST and COMSOL. A.S.A. designed and fabricated the devices. A.H. and J.F. wrote the manuscript with help from other co-authors. The work was done under the supervision of M.L. and J.F.

## Competing interests
The authors declare no competing interests.
