## [Transparent Peer Review file · Nature Communications]

Thin-film lithium niobate terahertz differential field detectors with a bandwidth reaching 3 terahertz

Corresponding Author: Ms Alexa Herter

Version 0:

Reviewer comments:

Reviewer #1

(Remarks to the Author)

Recently, Thin Film Lithium Niobate has attracted much attention as an integrated photonics device, and many related papers have been published. Not only in the field of telecommunication applications, but also in various sensing applications, many applications have been reported using integrated platforms based on the nonlinearity of TFLN instead of conventional Si waveguides.

As examples, a paper on a broadband spectrometer on a TFLN platform [1] was published in Nature Photonics, and a paper using a Mach-Zehnder circuit on TFLN as an interaction field between millimeter waves and light waves [2] was reported as a paper related to this review paper.

The author of this paper has also published "Terahertz waveform synthesis in an integrated thin-film lithium niobate platform"[3] in this journal.

Then, this paper was reviewed with taking account of the published papers like above. First, the paper addresses the content of a proposed optical Mach-Zehnder circuit with a THz-wave antenna configured on a TFLN to perform broadband THz-wave detection using the non-resonant nature of the interaction. The paper also shows that an integrated Mach-Zehnder interferometer with two antennas can be used to measure the discrete-time derivative of the THz waveform, and that a single antenna can be used to measure the instantaneous derivative of the electric field.

While the results are interesting from a general point of view, the report is positioned as an extension or continuation of the authors' work on "Terahertz waveform synthesis in integrated thin-film lithium niobate platform" [3]. And also, it is characterized by the fact that technical contents are mainly reported. From a general viewpoint, the authors have already published about the related paper in this journal, then it is expected more impact, progressiveness, innovativeness, and exceptional results beyond the previous paper by the authors. On the other hand, the authors state that this achievement will realize the next generation of compact detectors that can be applied to spectroscopy and quantum optics. If so, it is evaluated that the effectiveness and progress of this achievement must be concretely included in this paper as demonstration of the device in such applications.

Reviewer #2

(Remarks to the Author)

The reviewed work demonstrates terahertz (THz) field detector with the detection principles based on the non-linearity of lithium niobate affected by the antenna enhanced electric field. Two solutions are introduced: combination of two spatially separated antennas in a Mach-Zehnder interferometer and single antenna. I consider the article to be well-prepared and expected to be interesting not only for the specialists of the related field, but also for broad scientific community beyond. However, I would like to raise several minor considerations/suggestions in order to enhance the article and/or remove misunderstanding.

1) Recognizing big experience of the authors in the presented field of study, I could not help but notice a considerably large amount of the same group citations in the reference list (46%). I do recognize the majority of the referenced works to be important for the particular research, however I would strongly recommend the authors to consider addition of the same-field achievements of other groups. For example, [I. Wilke et al., Scientific Reports, 14:4822 (2024)] or [J. Rollinson et al., JOSA B, 38:2 (2021)] from my point of view, can be considered to be quite closely related.

2) I would suggest the authors to reconsider (reword) the introduction part to better highlight the novelty of the particular research. The THz detection using the principles of Mach-Zehnder interferometry (different antenna and phase shifter) is revealed in [16]. The possibility of THz detection with single metallic bow-tie antenna (still PPS) is revealed in [17]. The same principles (electro-optic coupling) in the bow-tie + lithium niobate waveguide + Mach-Zehnder interferometry are exposed in [19] (generation). Finally, detection via the same as in-manuscript-presented single antenna + lithium niobate waveguide + Mach-Zehnder interferometry, wherein each arm comprises several antennas with spatial shift, is presented in [21]. This comment is mostly made not to question the novelty of the presented research, but more as a suggestion to consider introducing of a more clear highlight on the results presented within the particular research.

3) Continuing the previous, the reference [21], comprising the archiv.org version of supposedly submitted article, in my consideration looks more like a next step, succeeding the presented manuscript, resulting, as claimed in the reviewed manuscript, in „outstanding sensitivity and directionality“. According to the authors, the in-manuscript proposed solution is expected „to maximize the THz detection bandwidth“, however this comparison, although obvious, is neither performed or mentioned further. Authors are suggested to review the importance of the inclusion of reference [21] and/or including the comparison highlighting the differences with the cited research.

4) The geometrical parameters of the antenna (except for the arm length) are not given. Reader is referenced to [19], but the exact parameter set is unclear. Such data is expected to enhance the manuscript, considering the reproducibility aspects as well as for direct comparison possibility with the previous works.

5) Is the „relative modulation“ [1/THz] appearing in Fig. 3, Fig. 4a, and corresponding part of manuscript related to the relative modulation [dimensionless] being Δ/I (also consider adding units to the corresponding values in manuscript)? Authors are suggested to unify the terms in the text, figures, and equations. Also authors are suggested to comment on not including the modulation efficiency η as a parameter, which was successfully employed during previous research stages. Finally, authors are suggested to compare the performance of the presented detector with the ones presented before.

6) Authors are expected to clarify the method for the bandwidth estimation of the presented detector. This aspect is considered important as the operation on high frequency is placed in the title, thus clarity is expected. Currently in the manuscript it may seem that the cut-off frequency was estimated by „we see the measured signal“ principle. Suggestion would be to expand or employ classical -3dB or 20dB SNR rule, or any other to the preference of the authors.

7) Authors are suggested to include small discussion on the detection limits (in terms of frequency) of the presented approach and conditions for the limits to be achieved.

8) Investigation on the variation of the probe power (Figure 4) is performed on the device comprising 90 μm (not the one operating up to 3 THz as claimed). Do the observed dependencies remain in the case of device comprising smaller 40 μm antenna?

9) In the A part of the results antennas with resonance at 0.22 THz, 0.42 THz, and 1.03 THz are presented. In the B part, however, employment of "single antenna with a resonance frequency of 0.9 THz" is claimed. Is this a mistype and if not what is the reasoning for comparison of single antenna and interferometer-based devices as is done in Figure 5g?

10) In the article several effects are explained by the deviation of the operation point from the ideal. The solution is also proposed in conclusion section for the further research. But from the current work perspective, having mastered the simulation part, is it possible to obtain the exact value of deviation by combining experimental results and modeling?

Small typos and suggestions:

*) Abstract. Consider rewording: "We present a compact and high performance THz field detector based **on resonant antennas on a thin-film lithium niobate waveguides**".

*) Abstract. Typo: We show that combining two of such antennas in an integrated Mach Zehnder **interferometer** allows for a measurement of the discrete time derivative of the THz waveform while using a single antenna measures the instantaneous derivative of the field.

*) Introduction para 1. Consider rewording: spectroscopy mentioned twice in the same context.

*) Introduction para 7. Consider rewording: These two approaches not only simplify the fabrication process but also open new possibilities for quantum metrology, where knowledge of both the field **and with** its derivative is of great interest since they constitute two conjugate counterparts [11].

*) Figure 1. a: Bracket "]" is cutted out. c: Abbreviation MZI (Mach-Zehnder Interferometer) does not appear in text. d: Unrequired double brackets.

*) Figure 1. c: Marking the overall phase difference defined as $\phi_{\text{MZI}} + \phi_1 - \phi_2$ and employment of the $\phi_{\text{MZI}} = \pi/2$ marking on the

axis is considered misleading.

*) Results A, para 3. Extra space: "... index of lithium niobate at the probe **frequency** , $r_{33} = 35 \text{ pm V}^{-1}$ is the electro-optic coefficient of lithium niobate, ... "

*) Results A, para 11. Comma is missing: "At the respective resonance frequency of the antenna $\Omega_{\text{res}/2\pi=0.22 \text{ THz}}$, **0.42 THz** and 1.03 THz, the modeled responsivity".

*) Figure 2, caption. Capital letter missing: "to modeled signal. **the** measured relative transmission change $\Delta I/I$ is plotted as a function of time delay ..."

*) Results B, para 4. Consider changing to "an": "Since the signal is caused by the temporal derivation of a field, intuitively one expects **a** increasing signal for higher frequencies."

*) Results B, para 4. Consider rewording: "In both cases the spectral shape is **main** determined by the utilized antenna resonance, so that a high similarity in the spectral properties is observed."

*) Figure 3. Authors are suggested to include arm length value to the caption of the figure in order to increase clarity.

*) Figure 5. Some of the indexes are left inclined.

*) Conclusions. Consider changing to "highlights": "This **highlight** the broadband nature of the antenna's performance."

*) Supplementary. Please fix references.

*) Supplementary, equation 16. Typo: " $1+1\cos(\varphi_{\text{MZI}})$...".

*) Supplementary, caption of Figure 3. Please check the manufacturers name: "We are using a dual-wavelength laser system provided by **MenoloSystem** emitting femtosecond pulses origin from a erbium doped fiber ... "

*) Authors are suggested to introduce non-breaking spaces/dashes to simply succeeding publishing steps.

Reviewer #3

(Remarks to the Author)

The manuscript presents an intriguing technique for THz field detection using a thin-film lithium niobate waveguide platform. The authors describe two device configurations: (1) an interferometric setup with two THz antennas enabling direct THz field detection and (2) a single-waveguide, single-antenna device enabling the detection of the derivative of the THz field. The study highlights several novel physical aspects and potential applications, making it a valuable contribution to the field. I recommend publication in Nature Communications, but suggest the following revisions:

1. The manuscript lacks clarity on how the THz field detection specifications are quantified. For instance, it states: "We have achieved a noise-equivalent intra-cavity field as low as 1.9 V/m for an integration time of 100 ms, corresponding to a single-shot noise-equivalent field of 4.6 kV/m." Additionally, Fig. 4c presents the noise-equivalent field of power fluctuations detected with a 100 ms integration time per point, in the absence of any THz signal, as a function of probe power. When referring to field, do the authors mean the THz field or a single/differential DC field applied to the lithium niobate waveguide? How are these values measured, calibrated, and related to the device's responsivity to the THz field? Have separate measurements been conducted to determine the incident THz field when measuring the device response?
2. I suggest benchmarking the results against other THz field detectors, particularly photoconductive and electro-optic THz detectors, in terms of SNR when responding to the same pulsed THz source. Since noise-equivalent field is not commonly used to quantify the sensitivity of THz field detectors, a direct comparison with state-of-the-art technologies would provide better context for the reader.
3. The interferometric THz field detection approach appears to be sensitive to the direction of incoming THz radiation, given that the interferometer arm spacing is on the order of the THz wavelength. Please discuss this angular dependence.
4. While the manuscript accounts for the optical probe pulse width in its derivations, I recommend adding a discussion on how optical pulse width and dispersion affect the temporal resolution and frequency span of the detected THz field.
5. The comparison with a ZnTe electro-optic detector in Fig. 3 suggests a higher noise level for the TFLN device. Please provide an explanation for this (were different photodetectors used in these two measurements?).
6. Can the authors extract the THz field from its derivative, enabling the same device to be used for both direct THz field detection and derivative-based detection?

Version 1:

Reviewer comments:

Reviewer #1

(Remarks to the Author)

It is acknowledged that the authors have conducted excellent research that is leading the field. Some points have been made clearer through the revision of the paper. On the other hand, there are some aspects that I would like to clear up as novel as the content of our contribution to this journal. As stated in the authors' response letter, I understand that detection and generation of THz waves are not necessarily complementary. On the other hand, the authors argue that introducing the new capability of high sensitivity, broadband, on-chip detection of THz signals using a non-resonant MZI architecture is a complementary and necessary step toward a complete THz system-on-chip solution. Then, an important point of this paper is to understand how the authors' group intends to leapfrog from their already published work on THz wave detection [ref. 26]. Based on the research results already published by the group, the introduction should make it clear that this paper is trying to find the future potential of this pioneering and advanced technology that will lead to the next step. Since the group has already published a paper, this is a continuing point of discussion within the group.

Ref. 26

A. Tomasino, A Shams-Ansari, M. Lončar, I.-C. Benea-Chelmus, "Large-area photonic circuits for terahertz detection and beam profiling, " (2024).
<https://doi.org/10.48550/arXiv.2410.20407>

Deployment of terahertz communication and spectroscopy systems relies on the availability of low-noise and fast detectors, with plug-and-play capabilities. However, most currently available technologies are stand-alone, discrete components, either slow or susceptible to temperature drifts. Moreover, phase-sensitive schemes are mainly based on bulk crystals and require tight beam focusing. Here, we demonstrate an integrated photonic architecture in thin-film lithium niobate that addresses these challenges by exploiting the electro-optic modulation induced by a terahertz signal onto an optical beam at telecom frequencies. Leveraging on the low optical losses provided by this platform, we integrate a double array of up to 18 terahertz antennas within a Mach-Zehnder interferometer, considerably extending the device collection area and boosting the interaction efficiency between the terahertz signal and the optical beam. We show that the double array coherently builds up the probe modulation through a mechanism of quasi-phase-matching, driven by a periodic terahertz near-field pattern, without physical inversion of the crystallographic domains. The array periodicity controls the detection bandwidth and its central frequency, while the large detection area ensures correct operation with diverse terahertz beam settings. Furthermore, we show that the antennas act as pixels that allow reconstruction of the terahertz beam profile impinging on the detector area. Our on-chip design in thin-film lithium niobate overcomes the detrimental effects of two-photon absorption and fixed phase-matching conditions, which have plagued previously explored electro-optic detection systems, especially in the telecom band, paving the way for more advanced on-chip terahertz systems.

Reviewer #2

(Remarks to the Author)

In my opinion, the authors made a significant effort to address the comments of all of the Reviewers', which led to a substantial improvement of the article. Considering the answers provided and the resubmitted version of the manuscript I believe the manuscript can be published as is.

Reviewer #3

(Remarks to the Author)

My questions and comments are all addressed.

Version 2:

Reviewer comments:

Reviewer #1

(Remarks to the Author)

The authors have made it easier to grasp the significance of their previous research and to understand the novelty of the advancements made beyond their earlier publications.

Response to referees

We would like to thank all reviewers for their positive assessment of our work. In detail, we would like to thank referee 1 for summarizing our achievements in the field of integrated THz technologies, referee 2 for considering our findings interesting for both, specialists but also a broad scientific community, and referee 3 for seeing our work as a valuable contribution to the field.

Given the suggestions and comments we received in the first round of reviews, we have decided to undertake the following major revisions to the initially submitted version of the manuscript and supplementary information:

1. Rewording large parts of the introduction and the discussion of the noise-properties quantified for the interferometric detection scheme.
2. Extended the conclusion by an extensive discussion about the upper frequency limitations of the presented detection scheme.
3. Added a comparison of the expected and measured signal amplitude for a set of different devices to estimate the uncertainty of the phase difference between the two interferometer arms to the Supplementary material.

Furthermore, we made the following small revisions to the manuscript:

1. Decouple the definition of the electro-optic coupling strength g_{eo} from the overlap factor Γ to be independent from the choice of normalization. The normalization of Γ is linked to the normalization of the field-enhancement $G(\Omega)$ and it's value is not independent.
2. Added a table to the Supplementary material listing all relevant parameters of the different device designs investigated in the study.

In the following, the questions by the referees are shown in blue, answers are shown in black and changes to the manuscript are shown in green.

1 Reviewer 1

1. Recently, Thin Film Lithium Niobate has attracted much attention as an integrated photonics device, and many related papers have been published. Not only in the field of telecommunication applications, but also in various sensing applications, many applications have been reported using integrated platforms based on the nonlinearity of TFLN instead of conventional Si waveguides. As examples, a paper on a broadband spectrometer on a TFLN platform [1] was published in Nature Photonics, and a paper using a Mach-Zehnder circuit on TFLN as an interaction field between millimeter waves and light waves [2] was reported as a paper related to this review paper. The author of this paper has also published "Terahertz waveform synthesis in an integrated thin-film lithium niobate platform"[3] in this journal. Then, this paper was reviewed with taking account of the published papers like above. First, the paper addresses the content of a proposed optical Mach-Zehnder circuit with a THz-wave antenna configured on a TFLN to perform broadband THz-wave detection using the non-resonant nature of the interaction. The paper also shows that an integrated Mach-Zehnder interferometer with two antennas can be used to measure the discrete-time derivative of the THz waveform, and that a single antenna can be used to measure the instantaneous derivative of the electric field. While the results are interesting from a general point of view, the report is positioned as an extension or continuation of the authors' work on "Terahertz waveform synthesis in integrated thin-film lithium niobate platform" [3]. And also, it is characterized by the fact that technical contents are mainly reported. From a general viewpoint, the authors have already published about the related paper in this journal, then it is expected more impact, progressiveness, innovativeness, and exceptional results beyond the previous paper by the authors. On the other hand, the authors state that this achievement will realize the next generation of compact detectors that can be applied to spectroscopy and quantum optics. If so, it is evaluated that the effectiveness and progress of this achievement must be concretely included in this paper as demonstration of the device in such applications.

- [1] Pohl, D., Reig Escalé, M., Madi, M. et al. "An integrated broadband spectrometer on thin-film lithium niobate", Nat. Photonics 14, 24–29 (2020). <https://doi.org/10.1038/s41566-019-0529-9>
- [2] Yiwen Zhang, Linbo Shao, Jingwei Yang, Zhaoxi Chen, Ke Zhang, Kam-Man Shum, Di Zhu, Chi Hou Chan, Marko Lončar, and Cheng Wang, "Systematic investigation of millimeter-wave optic modulation performance in thin-film lithium niobate," Photon. Res. 10, 2380-2387 (2022)
- [3] Herter, A., Shams-Ansari, A., Settembrini, F.F. et al. "Terahertz waveform synthesis in integrated thin-film lithium niobate platform", Nat. Commun. 14, 11 (2023). <https://doi.org/10.1038/s41467-022-35517-6>

The authors thank the reviewer for the comments and the summary of our work. We respectfully disagree with the reviewer's impression that this work may appear as an extension of our previous publication on THz waveform synthesis. We want to stress that detection and generation are fundamentally distinct domains, each with its own set of challenges, techniques, and significance. This work introduces a new capability: high-sensitivity, broadband, on-chip detection of THz signals using a non-resonant MZI architecture, which is a complementary and necessary step toward enabling full THz system-on-chip solutions. We hope that this work lays the foundation for a growing subfield in which other groups will pursue advanced detection architectures just as others have done in more common wavelength regions such as telecommunications or near-infrared. We give a proof of principle for the integrated detection principle on thin-film lithium niobate and its characterization, while the further investigation and development required for the application in THz spectroscopy or quantum sensing will be the subject of future studies. For more clarity on the distinction from our previous work and the future steps, we revised the Introduction (see also Question 2 of Reviewer 2) and as a result, the conclusion now better highlights potential application of our new device.

2 Reviewer 2

1. Recognizing big experience of the authors in the presented field of study, I could not help but notice a considerably large amount of the same group citations in the reference list (46%). I do recognize the majority of the referenced works to be important for the particular research, however I would strongly recommend the authors to consider addition of the same field achievements of other groups. For example, [I. Wilke et al., Scientific Reports, 14:4822 (2024)] or [J. Rollinson et al., JOSA B, 38:2 (2021)] from my point of view, can be considered to be quite closely related.

We thank the reviewer for this comment and fully agree that the reference list lacks the achievements of other researchers. In addition to the two works mentioned by reviewer 2, we discuss more references outside our own research groups, e.g., about photoconductive antennas, integrated spectrometer, and high-frequency modulators on thin-film lithium niobate in the introduction (cf. the new version of the manuscript).

2. I would suggest the authors to reconsider (reword) the introduction part to better highlight the novelty of the particular research. The THz detection using the principles of Mach-Zehnder interferometry (different antenna and phase shifter) is revealed in [16]. The possibility of THz detection with single metallic bow-tie antenna (still PPS) is revealed in [17]. The same principles (electro-optic coupling) in the bow-tie + lithium niobate waveguide + Mach-Zehnder interferometry are exposed in [19] (generation). Finally, detection via the same an in-manuscript-presented single antenna + lithium niobate waveguide + Mach-Zehnder interferometry, wherein each arm comprises several antennas with spatial shift, is presented in [21]. This comment is mostly made not to question the novelty of the presented research, but more as a suggestion to consider introducing of a more clear highlight on the results presented within the particular research.

The authors thank the reviewer for the helpful suggestion. Following this we reviewed and restructured the paragraph in the introduction on the progress in integrated THz photonics and the novelty of our work (cf. the new version of the manuscript).

3. Continuing the previous, the reference [21], comprising the [archiv.org](https://www.archiv.org) version of supposedly submitted article, in my consideration looks more like a next step, succeeding the presented manuscript, resulting, as claimed in the reviewed manuscript, in “outstanding sensitivity and directionality”. According to the authors, the in-manuscript proposed solution is expected “to maximize the THz detection bandwidth”, however this comparison, although obvious, is neither performed or mentioned further. Authors are suggested to review the importance of the inclusion of reference [21] and/or including the comparison highlighting the differences with the cited research.

The authors thank the reviewer for this comment. Because it utilizes the same technological platform and also concerns the detection of THz waves, we believe that the reference [21] should be cited. However, we have now better highlighted the clear differences between this work and the one presented here, and as such the text has been modified:

“Recently, the combination of an integrated MZI structure on TFLN with arrays of THz antennas has demonstrated narrowband THz detection, emphasizing high sensitivity and spatial resolution across large-area detectors. In contrast, the present work focuses on broadband THz detection and a detailed description of two different detection principles to understand the potential for further application. The derived signal in the frequency domain of the interferometric devices can also be directly applied to quantum states of light in the THz range. We offer a fundamental integrated element that can be further extended or combined with other on-chip technologies.”

4. The geometrical parameters of the antenna (except for the arm length) are not given. Reader is referred to [19], but the exact parameter set is unclear. Such data is expected to enhance the manuscript, considering the reproducibility aspects as well as for direct comparison possibility with the previous works.

The authors thank the reviewer for pointing out the lack of clarity. We added a table to the Supplementary Material listing all design parameters differing between the investigated devices and the simulated resonance frequency of the particular antenna design and refer to it in the beginning of the Results section:

“All the devices are fabricated on a *x*-cut thin-film lithium niobate chip (details about fabrication and the photonic chip in Methods section and Supplementary material of [1], design parameters varying between the devices shown in this study are listed in Tab. 1 of the Supplementary Material).”

5. Is the “relative modulation” [1/THz] appearing in Fig. 3, Fig. 4a, and corresponding part of manuscript related to the relative modulation [dimensionless] being $\Delta I/I$ (also consider adding units to the corresponding values in manuscript)? Authors are suggested to unify the terms in the text, figures, and equations. Also authors are suggested to comment on not including the modulation efficiency η as a parameter, which was successfully employed during previous research stages. Finally, authors are suggested

to compare the performance of the presented detector with the ones presented before.

We thank the reviewer to mention the lack of clarity. In the time-domain the relative modulation $\Delta I(\tau)/I$ is indeed dimensionless. When applying the Fourier transformation, the spectral relative modulation $\Delta I(\Omega)/I$ will have the unit of 1 per frequency. For more clarity, we now specify throughout the manuscript whether the time-dependent relative modulation $\Delta I(\tau)/I$ or the spectral relative modulation $\Delta I(\Omega)/I$ is considered.

We decided to not include the modulation efficiency, which is commonly used to describe the performance of modulators, because of its dependency on the interaction/modulator length and therefore the THz frequency of interest. We have now introduced the relative modulation per electric field and interaction length at the antenna resonance frequency assuming perfect phase-matching and infinitesimal short pulses to quantify the specific capability to modulate the near-infrared signal independent of the properties of the utilized probe pulse and the technique to optimize for phase-matching. The following comparison to free-space electro-optic sampling has been added to the text:

“The modulation per incident THz field and interaction length $\eta_{\text{LN}} = \frac{n_{\text{p}}^3 \omega_{\text{p}} r_{33}}{c} \cdot \Gamma \cdot G(\Omega)$ at the resonance frequency of the particular antenna is $\eta_{\text{LN}} = 80 \times 10^{-3} \text{ V}^{-1}$, $29 \times 10^{-3} \text{ V}^{-1}$ and $22 \times 10^{-3} \text{ V}^{-1}$ for the three devices at 0.22 THz, 0.42 THz and 1.03 THz, respectively. In comparison, the corresponding value for electro-optic sampling $\eta_{\text{EOS}} = \frac{n^3 \omega_{\text{p}} r_{41}}{c}$ is about two orders lower and measures $1.16 \times 10^{-3} \text{ V}^{-1}$ using zinc telluride and a probe wavelength of 800 nm or $0.23 \times 10^{-3} \text{ V}^{-1}$ in gallium arsenite with a probe signal at 1560 nm.”

6. Authors are expected to clarify the method for the bandwidth estimation of the presented detector. This aspect is considered important as the operation on high frequency is placed in the title, thus clarity is expected. Currently in the manuscript it may seem that the cut-off frequency was estimated by “we see the measured signal” principle. Suggestion would be to expand or employ classical -3 dB or 20 dB SNR rule, or any other to the preference of the authors.

THz time-domain spectroscopy systems take advantage of the very large dynamical range of the signal measured to enable useful measurements to be performed much lower than the 3dB cutoff of the signal. For this reason, we followed the convention of the field and called the “bandwidth” the signal above some noise, as it is done both in the literature and in the industry (cf. Toptica, [2, 3]). To achieve more clarity in the text on the definition of bandwidth in this field, we added the following to the manuscript:

“Although the simulated antenna resonance is at 1.03 THz the observable signal reaches 3 THz, which defines the bandwidth in the context of terahertz time-domain spectroscopy.”

7. Authors are suggested to include small discussion on the detection limits (in terms of frequency) of the presented approach and conditions for the limits to be achieved.

We thank the Reviewer for this suggested and added the following discussion about limitations in the detection bandwidth to the conclusion of the manuscript:

“In the study of the smallest antenna optimized for detection at 1 THz, we observed a signal extending to 3 THz even with a relatively long pulse duration of approximately 400 fs, which determines the time resolution similar to other time-resolved THz detection schemes. Electro-optic sampling in bulk crystal demonstrated, that the frequency-limitation caused by the probe duration can be extended into the mid-infrared frequency range, when using sub-10 fs pulses [4]. For a fiber-coupled system using the TFLN chip as shown in this study, dispersion-complementing fibers are widely available for the probe signal at the telecommunication wavelength. However, also the design of LN waveguides allows one to control the dispersion, so that for a fully integrated solution in the future development, Fourier-limited pulses in the detection region could be achieved. While the bow-tie antenna design used in the current study shows a broadband nature in the lower frequency range, different antenna designs as e.g. split-ring resonators can be used to optimize for detection at higher THz frequencies. In the current work, the comparable short interaction length ensures efficient phase-matching, even when assuming a standing THz-wave. At the same time, the integrated platform offers the advantage of adapting the propagation speed of the THz and probe signal by the design of the antenna geometry, the interaction region formed by the gold bars in its gap and the near-infrared waveguide, which has already been successfully applied to the generation of THz radiation in thin-film lithium niobate [5]. The phonon resonance at 4.5 THz limits the

detection in bulk lithium niobate to below 2 THz. In contrast, the bandwidth of the integrated detection exceeds this limitation because of the low interaction of the THz radiation with the only 300 nm-thick lithium niobate layer. In fact, the usage of a lithium niobate waveguide in free-space electro-optic sampling has demonstrated a significant increase in the detection bandwidth [6]. The mentioned study predicts a detection bandwidth of up to 6.5 THz, which underlies the capacity of the THz detection on thin-film lithium niobate reaching a much higher bandwidth than shown so far.”

8. Investigation on the variation of the probe power (Figure 4) is performed on the device comprising 90 μm (not the one operating up to 3 THz as claimed). Do the observed dependencies remain in the case of device comprising smaller 40 μm antenna?

The authors thank the reviewer for this comment and agree that a smaller antenna design would be better suited to discuss the spectral response depending on the probe power used. Unfortunately, the power dependency has only been measured with the 90 μm -device. However, we do not expect any differences in the observed dependencies, since the behavior of the observed noise is independent of the frequency of the detected signal. If the signal is strongly affected by higher-order nonlinear effects disturbing the pulse-shape of the probe signal, we would expect to see changes in the detector also at the frequencies covered by the 90 μm -device. We added:

“A reproducible THz spectrum is observed within the full power range (Fig. 5 a), which indicates that nonlinear effects in the on-chip waveguide do not affect the spectral response of this device within the investigated power range.”

to clarify that our conclusions are limited to the spectral range of the investigated device.

9. In the A part of the results antennas with resonance at 0.22 THz, 0.42 THz, and 1.03 THz are presented. In the B part, however, employment of ”single antenna with a resonance frequency of 0.9 THz” is claimed. Is this a mistype and if not what is the reasoning for comparison of single antenna and interferometer-based devices as is done in Figure 5g?

The authors thank the reviewer for pointing out the lack of clarity. For the current study, devices with different dimensions were fabricated. In addition to the arm length of the antenna, the arm width, the gap length and width have also been modified, which also influence the resonance properties of the antenna. In part A we choose for the 40 μm -device the design with the highest frequency, while in part B, we choose a combination of Mach-Zehnder interferometer and a single antenna with parameters as similar as possible. Since the two device-types belong to different fabrication generations, we do not have two devices with identical antenna parameters.

For more clarification, we added a table to the Supplementary material listing the design parameters of all devices investigated in this study including the simulated resonance frequencies and refer to it at the beginning of the results section (see also Reviewer 2, Question 4) and when investigating the single antenna device:

“To verify the theoretical prediction, we now use an integrated device consisting of a single antenna with a resonance frequency around 0.9 THz to measure the THz emission of the previously utilized photoconductive antenna (antenna parameters in the Supplementary Material Tab. 1).”.

10. In the article several effects are explained by the deviation of the operation point from the ideal. The solution is also proposed in conclusion section for the further research. But from the current work perspective, having mastered the simulation part, is it possible to obtain the exact value of deviation by combining experimental results and modeling?

The authors thank the reviewer for this question. Inspired by this question, we compared a set of measurement using different devices to the expected signal calculated by the presented model. From the ratio between the measured and simulated values we can conclude on the corresponding phase-difference between the two interferometer arms, that would lead to this deviation. However also other uncertainties like the exact temporal shape of the probe in the interaction region, the alignment, and the true electric field focused onto the device in this setup can affect the signal amplitude as well. Therefore, these values are not exact, but the phase-difference seems to cover the range from $\sim 0.25\pi$ to $\sim 0.7\pi$. We have now added the mentioned analysis to the Supplementary material and add to the main text:

“Measuring 10 different devices under the same conditions, the observed amplitude varies between 50 % and 2.5 times the modeled signal. Since other uncertainties such as the true value of the THz electric field incident on the sample or the exact temporal shape of the probe pulses would cause a systematic deviation,

we attribute the observed variation to the operation point of the interferometer. The observed deviation range corresponds to the phase-difference varying between 0.25π and 0.7π (details in the Supplementary Materials Section 3.6.”

11. The authors thank the reviewer for his detailed feedback on our work. We implemented all the further mentioned suggestions about typos, bad wording, and inconsistencies.

3 Reviewer 3

1. The manuscript lacks clarity on how the THz field detection specifications are quantified. For instance, it states: ”We have achieved a noise-equivalent intra-cavity field as low as 1.9 V/m for an integration time of 100 ms, corresponding to a single-shot noise-equivalent field of 4.6 kV/m.” Additionally, Fig. 4c presents the noise-equivalent field of power fluctuations detected with a 100 ms integration time per point, in the absence of any THz signal, as a function of probe power. When referring to field, do the authors mean the THz field or a single/differential DC field applied to the lithium niobate waveguide? How are these values measured, calibrated, and related to the device’s responsivity to the THz field? Have separate measurements been conducted to determine the incident THz field when measuring the response of the device?

We thank the reviewer for pointing out the lack of clarity and apologize for the misleading formulation of “noise-equivalent field of power fluctuations”. The noise-equivalent field is defined in analogy to the noise-equivalent power of an optical power detector. For more clarity, we have added its definition and re-worded the corresponding paragraph:

“In analogy to the noise-equivalent power of a power detector, we additionally determine the noise-equivalent field $NEF = \frac{1}{g_{eo}} \cdot \frac{\delta I_{std}}{I_0}$ by calculating the standard deviation of the detected photo-current fluctuations δI_{std} on top of the averaged photo-current I_0 in the absence of any coherent THz radiation or field along the waveguide. The noise-equivalent field is defined with respect to the field within the interaction region, consistent with quantum optics conventions, where field operators are referenced to confined modes rather than incident free-space fields. Typically, the sensitivity of electro-optic detection schemes is limited by the shot-noise in the photo-detectors $\frac{\delta I_{sn}}{I_0} = \sqrt{\frac{2e\Delta\nu}{I_0}}$, which is determined by the averaged induced photo-current I_0 and the detection bandwidth $\Delta\nu$ related to the chosen integration time of 100 ms in the current case. Fig. 4c demonstrates a similar inverse square-root relationship of the expected noise equivalent field $NEF_{sn} = \frac{1}{2g_{eo}} \cdot \frac{\delta I_{sn}}{I_0}$ caused by the shot-noise within almost the full power range. The lowest noise-equivalent field of 1.9 V m^{-1} is achieved using $76 \mu\text{W}$ averaged probe power. This value corresponds to a noise-equivalent field of 4.6 kV m^{-1} acquiring the signal at the laser’s repetition rate of 60 MHz as required to study the fluctuating properties of quantum states [7, 8]. ”

2. I suggest benchmarking the results against other THz field detectors, particularly photoconductive and electro-optic THz detectors, in terms of SNR when responding to the same pulsed THz source. Since noise-equivalent field is not commonly used to quantify the sensitivity of THz field detectors, a direct comparison with state-of-the-art technologies would provide better context for the reader.

The authors thank the reviewer for this suggestion. The noise properties of a field detection scheme are in particular important for applications in quantum optical experiments. In this context, the noise-equivalent field is a commonly used value. For comparison, we now also plot the noise-equivalent field over probe power for electro-optic sampling using two standard materials:

“The experimentally observed noise-equivalent field inside the antenna matches that of electro-optic sampling with zinc telluride, the standard material used with 800 nm probe pulses in state-of-the-art free-space electro-optic sampling, which is indicated by the solid gray line in Fig. 4c. Using gallium arsenide, which is commonly employed for electro-optic sampling using a C-band laser as in our case, the expected shot-noise equivalent field is approximately three times higher. In comparison to the anticipated shot noise for our detection scheme, we observe a noise level that is approximately 1.5 times higher, predominantly due to the relative intensity noise of the laser and to the electric noise of the transimpedance amplifier, whose gain had to be changed as the power is scaled. Furthermore, an increase in observed signal fluctuations is evident for the highest probe power of $162 \mu\text{W}$. ”

The SNR, commonly used for THz spectroscopy or in general for the detection of coherent THz fields in the classical limit, strongly depends on the performance of the THz emitter used in the experiment. In

the current work, we focus on the detection scheme on TFLN, while the emitting counterpart has been demonstrated very recently and its performance is expected to continuously improve during the further development process. To focus on the detection performance for classical THz detection, we introduce now the relative modulation per incoming THz field and interaction length. This allows us to compare to other time-resolve field detection schemes independent of the properties of the probe pulses and the method to achieve phase-matching. We have added to the results part when presenting the devices using three different antenna designs:

“The modulation per incident THz field and interaction length $\eta_{\text{LN}} = \frac{n_{\text{p}}^3 \omega_{\text{p}} r_{33}}{c} \cdot \Gamma \cdot G(\Omega)$ at the resonance frequency of the particular antenna is $\eta_{\text{LN}} = 80 \times 10^{-3} \text{ V}^{-1}$, $29 \times 10^{-3} \text{ V}^{-1}$ and $22 \times 10^{-3} \text{ V}^{-1}$ for the three devices at 0.22 THz, 0.42 THz and 1.03 THz, respectively. In comparison, the corresponding value for electro-optic sampling $\eta_{\text{EOS}} = \frac{n_{\text{c}}^3 \omega_{\text{p}} r_{41}}{c}$ is about two orders lower and measures $1.16 \times 10^{-3} \text{ V}^{-1}$ using zinc telluride and a probe wavelength of 800 nm or $0.23 \times 10^{-3} \text{ V}^{-1}$ in gallium arsenite with a probe signal at 1560 nm.”

3. The interferometric THz field detection approach appears to be sensitive to the direction of incoming THz radiation, given that the interferometer arm spacing is on the order of the THz wavelength. Please discuss this angular dependence.

In the current experimental configuration, we can assume normal incident of the THz radiation, since the sample thickness does not exceed the Rayleigh length of the focused THz beam. Due to the high refractive index of the silicon substrate of 3.425, the incidence angle of the free-space THz field will have only a small influence on the propagation angle inside the sample. As a consequence, the detection performance will be mainly affected by the reflection of the THz radiation at the sample substrate, when the incident angle of the free-space THz signal changes. We added in the derivation of equation (1):

“The experimental configuration in combination with the high refractive index of the silicon substrate ensures the propagation of the THz signal inside the device perpendicular to the chip surface.”

4. While the manuscript accounts for the optical probe pulse width in its derivations, I recommend adding a discussion on how optical pulse width and dispersion affect the temporal resolution and frequency span of the detected THz field.

The authors thank the reviewer for this suggestion. We added a discussion on pulse length and dispersion to the conclusion of the manuscript (see reviewer 2, question 7) and added:

“Similar to bulk electro-optic sampling [9], the effect of the pulse duration is expressed in the spectral autocorrelation of the probe signal: ”

when introducing $C(\Omega)$.

5. The comparison with a ZnTe electro-optic detector in Fig. 3 suggests a higher noise level for the TFLN device. Please provide an explanation for this (were different photodetectors used in these two measurements?). We thank the reviewer for this question. In both measurements, the noise level is dominated by shot-noise. In the measurement using zinc-telluride, the probe power was about 50 times higher than for the measurement using the integrated detector, explaining the difference in the noise floor. For more clarity in the manuscript we added:

“The noise level, which is determined in both cases by shot-noise and therefore related to the optical power reaching the photo-detector, is higher for the integrated detector due to the lower probe power of 0.14 mW compared to 5.8 mW detecting in bulk zinc-telluride.”

to the discussion of the measurement in Fig. 3 and give the expression of shot-noise in the discussion of the noise characterization in Fig. 4:

“Typically, the sensitivity of electro-optic detection schemes is limited by the shot-noise in the photo-detectors $\frac{\delta I_{\text{sn}}}{I_0} = \sqrt{\frac{2e\Delta\nu}{I_0}}$, which is determined by the averaged induced photo-current I_0 and the detection bandwidth $\Delta\nu$ related to the chosen integration time of 100 ms in the current case.”

6. Can the authors extract the THz field from its derivative, enabling the same device to be used for both direct THz field detection and derivative-based detection?

The authors thank the reviewer for this question. For classical fields, the THz field can be extracted from its derivation. From equation (2) we can extract the spectral field $E_{\text{THz}}(\Omega)$ and the Fourier transformation without multiplying with the term D_{shift} will give the THz field itself in the time domain. For quantum

states, the two quadratures of the electro-magnetic field are uncorrelated making the detection of the derivative additionally to the field itself particularly interesting (c.f. [10]).

References

- [1] A. Herter, A. Shams-Ansari, F. F. Settembrini, H. K. Warner, J. Faist, M. Lončar, and I. C. Benea-Chelms. “Terahertz waveform synthesis in integrated thin-film lithium niobate platform.” *Nature Communications* 2023 14:1, **14**:1–9 (2023).
- [2] L. Desplanque, J. F. Lampin, and F. Mollot. “Generation and detection of terahertz pulses using post-process bonding of low-temperature-grown GaAs and AlGaAs.” *Applied Physics Letters*, **84**:2049–2051 (2004).
- [3] S. B. Bodrov, A. N. Stepanov, M. I. Bakunov, B. V. Shishkin, I. E. Ilyakov, and R. A. Akhmedzhanov. “Highly efficient optical-to-terahertz conversion in a sandwich structure with LiNbO₃ core.” *Optics Express*, **17**:1871–1879 (2009).
- [4] A. Sell, R. Scheu, A. Leitenstorfer, and R. Huber. “Field-resolved detection of phase-locked infrared transients from a compact Er:fiber system tunable between 55 and 107 THz.” *Applied Physics Letters*, **93**:251107 (2008).
- [5] Y. Lampert, A. Shams-Ansari, A. Gaier, A. Tomasino, S. Rajabali, L. Magalhaes, M. Lončar, and I.-C. Benea-Chelms. “Photonics-integrated terahertz transmission lines.” (2024).
- [6] S. Mine, G. Gandubert, J. E. Nkeck, X. Ropagnol, K. Murate, and F. Blanchard. “Broadband heterodyne electro-optic sampling using a lithium niobate ridge-waveguide.” *Applied Physics Express*, **17**:042001 (2024).
- [7] C. Riek, D. V. Seletskiy, A. S. Moskalenko, J. F. Schmidt, P. Krauspe, S. Eckart, S. Eggert, G. Burkard, and A. Leitenstorfer. “Direct sampling of electric-field vacuum fluctuations.” *Science*, **350** (2015).
- [8] I. C. Benea-Chelms, F. F. Settembrini, G. Scalari, and J. Faist. “Electric field correlation measurements on the electromagnetic vacuum state.” *Nature*, **568** (2019).
- [9] A. S. Moskalenko, C. Riek, D. V. Seletskiy, G. Burkard, and A. Leitenstorfer. “Paraxial Theory of Direct Electro-optic Sampling of the Quantum Vacuum.” *Physical Review Letters*, **115** (2015).
- [10] P. Sulzer, K. Oguchi, J. Huster, M. Kizmann, T. L. Guedes, A. Liehl, C. Beckh, A. S. Moskalenko, G. Burkard, D. V. Seletskiy, and A. Leitenstorfer. “Determination of the electric field and its Hilbert transform in femtosecond electro-optic sampling.” *Physical Review A*, **101** (2020).

Response to referees

We would like to thank all reviewers for their positive assessment of our work. Given the suggestions and comments we received in the second round of reviews, we have decided to undertake the following revisions to the previously submitted version of the manuscript and supplementary information:

1. Rewording a paragraph in the introduction clarifying the distinction to the previous work in reference [29]
2. Clarify the relevance of broadband detection for integrated THz TDS and the availability of integrated components for a integrated system in the conclusion.

Furthermore, we made the following small change to the manuscript:

1. add two links references in the introduction, that were missing unintentionally in the previous submission.

In the following, the questions by the referees are shown in **blue**, answers are shown in black and changes to the manuscript are shown in **green**.

1 Reviewer 1

1. It is acknowledged that the authors have conducted excellent research that is leading the field. Some points have been made clearer through the revision of the paper. On the other hand, there are some aspects that I would like to clear up as novel as the content of our contribution to this journal. As stated in the authors' response letter, I understand that detection and generation of THz waves are not necessarily complementary. On the other hand, the authors argue that introducing the new capability of high sensitivity, broadband, on-chip detection of THz signals using a non-resonant MZI architecture is a complementary and necessary step toward a complete THz system-on-chip solution. Then, an important point of this paper is to understand how the authors' group intends to leapfrog from their already published work on THz wave detection [ref. 26]. Based on the research results already published by the group, the introduction should make it clear that this paper is trying to find the future potential of this pioneering and advanced technology that will lead to the next step. Since the group has already published a paper, this is a continuing point of discussion within the group.

[26] A. Tomasino, A Shams-Ansari, M. Lončar, I.-C. Benea-Chelmus, "Large-area photonic circuits for terahertz detection and beam profiling, " (2024). <https://doi.org/10.48550/arXiv.2410.20407>

Deployment of terahertz communication and spectroscopy systems relies on the availability of low-noise and fast detectors, with plug-and-play capabilities. However, most currently available technologies are stand-alone, discrete components, either slow or susceptible to temperature drifts. Moreover, phase-sensitive schemes are mainly based on bulk crystals and require tight beam focusing. Here, we demonstrate an integrated photonic architecture in thin-film lithium niobate that addresses these challenges by exploiting the electro-optic modulation induced by a terahertz signal onto an optical beam at telecom frequencies. Leveraging on the low optical losses provided by this platform, we integrate a double array of up to 18 terahertz antennas within a Mach-Zehnder interferometer, considerably extending the device collection area and boosting the interaction efficiency between the terahertz signal and the optical beam. We show that the double array coherently builds up the probe modulation through a mechanism of quasi-phase-matching, driven by a periodic terahertz near-field pattern, without physical inversion of the crystallographic domains. The array periodicity controls the detection bandwidth and its central frequency, while the large detection area ensures correct operation with diverse terahertz beam settings. Furthermore, we show that the antennas act as pixels that allow reconstruction of the terahertz beam profile impinging on the detector area. Our on-chip design in thin-film lithium niobate overcomes the detrimental effects of two-photon absorption and fixed phase-matching conditions, which have plagued previously explored electro-optic detection systems, especially in the telecom band, paving the way for more advanced on-chip terahertz systems.

The authors thank the reviewer for the comments and clarified the advantages of the current work compared to ref. [26] by the following reworded paragraph in the introduction:

"In this work, we demonstrate that resonant terahertz antennas combined with Mach-Zehnder interferometers on thin-film lithium niobate waveguides enable highly sensitive detection of broadband THz fields. In contrast to a recent report employing a similar platform [26], we use a single resonant antenna per interferometer arm instead of a pair of geometrical arrays, each consisting of up to nine antennas. This design choice enables the detection of THz waveforms with bandwidths up to 3 THz. In addition to the interferometric waveguide-based detection scheme, we investigate a second approach that is sensitive to the temporal derivative of the THz field. We provide a detailed analysis of the underlying detection principles, spectral response, and noise characteristics to assess the potential of THz detection on thin-film lithium niobate for applications in spectroscopy and quantum sensing."

Furthermore we further discussed the potential of the current technology for integrated THz time-domain spectroscopy by adding the following text to the conclusion of the paper:

"Since the generation of THz radiation on thin-film lithium niobate has already been demonstrated [26,27], the present demonstration of broadband and sensitive detection completes the foundation for time-domain THz spectroscopy on the same integrated platform. In particular, the achieved detection bandwidth up to 3 THz significantly broadens the range of spectroscopically accessible phenomena compared to previously reported narrowband detection schemes [29]. This extended bandwidth enables the investigation of a wider set of elementary excitations, including low-energy vibrational, rotational, and electronic transitions occurring across the full THz spectrum. In combination with the existing toolbox of integrated components

on thin-film lithium niobate [34], such as on-chip pulsed laser sources, active control of near-infrared probe power, and integrated THz transmission lines [27], the necessary technological prerequisites for realizing fully integrated THz time-domain spectroscopy are now in place.”

2 Reviewer 2

1. In my opinion, the authors made a significant effort to address the comments of all of the Reviewers’, which led to a substantial improvement of the article. Considering the answers provided and the resubmitted version of the manuscript I believe the manuscript can be published as is.

We thank the reviewer for their helpful comments and their support for the submission of the revised manuscript.

3 Reviewer 3

1. My questions and comments are all addressed.

We thank the reviewer for their helpful comments and their support for the submission of the revised manuscript.